# Global Linear and Local Superlinear Convergence of IRLS for Non-Smooth Robust Regression

**Liangzu Peng**
Mathematical Institute for Data Science
Johns Hopkins University
lpeng25@jhu.edu

**Christian Kümmerle**
Department of Computer Science
University of North Carolina at Charlotte
kuemmerle@uncc.edu

**René Vidal**
Mathematical Institute for Data Science
Johns Hopkins University
rvidal@jhu.edu

## Abstract

We advance both the theory and practice of robust $\ell_p$-quasinorm regression for $p \in (0, 1]$ by using novel variants of *iteratively reweighted least-squares* (IRLS) to solve the underlying non-smooth problem. In the convex case, $p = 1$, we prove that this IRLS variant converges globally at a linear rate under a mild, deterministic condition on the feature matrix called the *stable range space property*. In the non-convex case, $p \in (0, 1)$, we prove that under a similar condition, IRLS converges locally to the global minimizer at a superlinear rate of order $2 - p$; the rate becomes quadratic as $p \to 0$. We showcase the proposed methods in three applications: real phase retrieval, regression without correspondences, and robust face restoration. The results show that (1) IRLS can handle a larger number of outliers than other methods, (2) it is faster than competing methods at the same level of accuracy, (3) it restores a sparsely corrupted face image with satisfactory visual quality. https://github.com/liangzu/IRLS-NeurIPS2022

## 1 Introduction

Given a feature matrix $\boldsymbol{A} \in \mathbb{R}^{m \times n}$ with $m \gg n$ and a response vector $\boldsymbol{y} \in \mathbb{R}^m$, the problem

$$\min_{\boldsymbol{x} \in \mathbb{R}^n} \big\| \boldsymbol{A}\boldsymbol{x} - \boldsymbol{y} \big\|_p \tag{1}$$

of $\ell_p$-regression has a variety of different applications depending on the choice of $\|\boldsymbol{v}\|_p = \big( \sum_i |\boldsymbol{v}_i|^p \big)^{1/p}$. While $p = 2$ corresponds to standard linear regression, the choice of $p > 2$ arises naturally in semi-supervised learning on graphs [1, 2, 3], and a lot of activity has been dedicated recently to the computational complexity analysis for the case $p > 1$ [4, 5, 6].

In this paper, we assume there is a coefficient vector $\boldsymbol{x}^* \in \mathbb{R}^n$ such that the residual $\boldsymbol{r}^* := \boldsymbol{A}\boldsymbol{x}^* - \boldsymbol{y}$ is $k$-sparse, in which case a choice of $p \in (0, 1]$ is of interest. Indeed, for the convex and non-smooth case $p = 1$, (1) is known as *least absolute deviation*, which dates back to the time of Boscovich around in the middle of the 18th century [7, 8]. Since then, it has been known intuitively that (1) ($p = 1$) is robust to *large but few measurement errors* (quoting [9]), i.e., that (1) is robust to outliers.

The algorithmic and theoretical understanding of (1) for $p = 1$ has been of long-standing interest to statisticians, which has led to a vast literature spanning from the classic 1964 paper of Huber [10] to recent contributions [11, 12, 13, 14, 15, 16, 17, 18, 19, 20]. That being said, for $p = 1$, there is

36th Conference on Neural Information Processing Systems (NeurIPS 2022).

**Algorithm 1:** IRLS for $\ell_p$-Regression ($\texttt{IRLS}_p$)

---

1 Input: $\boldsymbol{A} = [\boldsymbol{a}_1, \ldots, \boldsymbol{a}_m]^\top \in \mathbb{R}^{m \times n}, \boldsymbol{y} = [y_1, \ldots, y_m]^\top \in \mathbb{R}^m, p \in (0, 1]$;

2 Weight initialization $\boldsymbol{w}^{(0)} \leftarrow [w_1^{(0)}; \ldots; w_m^{(0)}] \in \mathbb{R}^m$; // Instead, one can initialize some vector $\boldsymbol{x}^{(1)}$.

3 For $t \leftarrow 0, 1, \ldots$ :
$$\boldsymbol{x}^{(t+1)} \leftarrow \underset{\boldsymbol{x} \in \mathbb{R}^n}{\operatorname{argmin}} \sum\nolimits_{i=1}^{m} w_i^{(t)} (\boldsymbol{a}_i^\top \boldsymbol{x} - y_i)^2 \tag{2}$$

Update $\epsilon^{(t+1)}$ suitably based on $\boldsymbol{x}^{(t+1)}$, $\boldsymbol{A}$, and $\boldsymbol{y}$; // See Section 2.2 for details.
$$w_i^{(t+1)} \leftarrow \max \left\{ |\boldsymbol{a}_i^\top \boldsymbol{x}^{(t+1)} - y_i|, \epsilon^{(t+1)} \right\}^{p-2} \quad \forall i = 1, \ldots, m \tag{3}$$

---

a close, but somewhat underexplored connection between (1) and the *basis pursuit* problem [21], and more specifically, the theoretical [22, 23, 24] and algorithmic aspects [25, 26, 27] of compressed sensing [28] (see also Section 2.1). For $p \in (0, 1)$, problem (1) is non-smooth and non-convex, and generally less well understood. To the best of our knowledge, the only paper that considered (1) with $p \in (0, 1)$ is [29], where the authors presented a condition (based on *restricted isometry constants*) that guarantees exact recovery of $\boldsymbol{x}^*$ from (1). Other related works come either from the compressed sensing literature, where an $\ell_p$ (and noisy) version of basis pursuit has been considered as a sparsity-promoting formulation [30, 31, 32, 26, 33, 34, 35, 36, 37], or from the literature of matrix recovery/completion, where the *Schatten-p* norm has come into play as a non-convex surrogate for rank minimization [38, 39, 40, 41, 42, 43]. A key message from these works is that $\ell_p$ or Schatten-$p$ minimization with $p \in (0, 1)$ offers better information-theoretic properties (e.g., requires fewer samples for exact recovery) than minimization with $p = 1$.

Here, we study an *iteratively reweighted least-squares* method ($\texttt{IRLS}_p$) to solve (1) with $p \in (0, 1]$. As listed in Algorithm 1, $\texttt{IRLS}_p$ alternates between solving a weighted least-squares problem (2) and updating the weights (3); see Section 2.2 for more elaboration on IRLS. The simplicity of this idea (with its impressive performance) justifies its popularity in many machine learning [44, 45, 46, 47] and computer vision [48, 49, 50, 51, 52] applications. Based on recent advances on IRLS [26, 27], we make several contributions for understanding (1) and $\texttt{IRLS}_p$. We state our contributions next.

**The Stable Range Space Property (Section 3.1).** We put forward the use of the (stable) *range space property* (RSP) for studying the robust regression problem (1). The stable RSP was proposed by [53] in a different context to analyze a compressed sensing algorithm for solving a weighted basis pursuit problem at each iteration. In analogy to the *nullspace property* in compressed sensing [24, 28], we show that the RSP is a necessary and sufficient condition for guaranteeing that the $\ell_p$-regression problem in (1) admits $\boldsymbol{x}^*$ as its unique solution (Proposition 2). Moreover, we show that if $\boldsymbol{A} \in \mathbb{R}^{m \times n}$ has i.i.d. $\mathcal{N}(0, 1)$ entries and if $m$ is large enough, then the (stable) RSP holds with high probability (Proposition 3). This justifies its use as the core assumption in our analysis.

**Global Linear Convergence (Section 3.2).** We prove in Theorem 1 that, under a stable RSP assumption and with $\epsilon^{(t)}$ suitably updated, the $\texttt{IRLS}_p$ Algorithm 1 with $p = 1$ converges linearly to the ground-truth $\boldsymbol{x}^*$ from any initial weight $\boldsymbol{w}^{(0)}$ (or equivalently from any initial point $\boldsymbol{x}^{(1)}$). Note that while (accelerated) first-order methods (e.g., (sub-)gradient descent and proximal algorithms) can also solve the convex and non-smooth problem (1) with $p = 1$, they exhibit, in general, (global) sub-linear rates at best [54, 55]. To our knowledge, [17] is the only paper that claims global linear convergence of a different IRLS variant for (1) with $p = 1$; we compare our results with the ones of [17] in Section 3.2. On the other hand, Theorem 1 is inspired by the IRLS method of [27] for basis pursuit; based on [27], we suitably modify their proof strategy, and thus obtain a faster linear rate.

**Local Superlinear Convergence (Section 3.3).** We prove in Theorem 2 that, under a stable RSP assumption and with $\epsilon^{(t)}$ suitably updated, the $\texttt{IRLS}_p$ Algorithm 1 with $p \in (0, 1)$ converges to $\boldsymbol{x}^*$ superlinearly, provided that $\boldsymbol{x}^{(1)}$ falls into a certain neighborhood of $\boldsymbol{x}^*$. To the best of our knowledge, no similar result exists for $\ell_p$-regression (1). While Theorem 2 is inspired by the IRLS algorithm of [26] for $\ell_p$-basis pursuit, their IRLS method does not work well for small $p$ (Section 2.2), and unlike in our work, their radius of local convergence diminishes greatly as $p \to 0$ or $m \to \infty$.

**Applications (Section 4).** We illustrate the performance of $\mathtt{IRLS}_p$ for *real phase retrieval* [56, 57], *linear regression without correspondences* [58, 59, 60, 61], and *face restoration from sparsely corrupted measurements* [62, 63, 64]. For real phase retrieval (Section 4.1), we show that $\mathtt{IRLS}_{0.1}$ needs only $m = 2n - 1$ measurements to recover $x^*$ up to sign, with an additional assumption. Theoretically, even brute-force can fail to identify $\pm x^*$ for fewer than $2n - 1$ measurements [65]. Empirically, many methods, including *Kaczmarz* [66, 67, 57], *PhaseLamp* [68], *truncated Wirtinger flow* [69], and *coordinate descent* [70], fail with $m = 2n - 1$ Gaussian measurements (Figure 2a). For linear regression without correspondences (Section 4.2), we show in Figures 2b-2c that, $\mathtt{IRLS}_{0.1}$ is uniformly faster (20-100x) and more accurate than *PDLP* [71] (merged into Google or-tools) and the commercial solver *Gurobi* [72], both of which solve (1) with $p = 1$ as a linear program, and also than *subgradient descent* implemented by Beck & Guttmann-Beck [73]. For face restoration (Section 4.3), we present both quantitative and qualitative results on the Extended Yale B dataset [74].

## 2 Background

### 2.1 Connection of Robust Regression and Compressed Sensing

The $\ell_p$-regression problem (1) has a natural correspondence to the sparse recovery problem

$$\min_{r \in \mathbb{R}^m} \|r\|_p \quad \text{s.t.} \quad Dr = z \tag{4}$$

where $D$ is a $(m - n) \times m$ matrix, $z \in \mathbb{R}^{m-n}$ a vector and the objective $\|r\|_p$ penalizes coefficient vectors with too many non-zero coordinates. For $p = 1$, (4) is also called *basis pursuit* [75]. Problems (1) and (4) are known to be related in the following sense (as implicitly stated in [21, 29]):

**Proposition 1.** *Suppose that the range space of $A$ is equal to the nullspace of $D$, and that $z = -Dy$. If $x_1$ globally minimizes (1), then $r_1 := Ax_1 - y$ globally minimizes (4). On the other hand, if $r_2$ globally minimizes (4), then there exists some $x_2$ with $r_2 = Ax_2 - y$ that globally minimizes (1).*

Proposition 1 sheds light on how we "transfer", with new insights, from the analysis of [26, 27] for (4) to results for (1). In what follows, we treat [26, 27] in the context of robust regression and highlight the contribution of our work relative to [26, 27] and other existing results, whenever possible.

### 2.2 Iteratively Reweighted Least-Squares: The Basics and New Insights

We first discuss two different updating rules for the *smoothing parameter* $\epsilon^{(t)}$ of Algorithm 1: the *fixed* rule and the *dynamic* rule. We then consider the weight updating strategy (3). Along the way we use synthetic experiments to illustrate ideas; see Appendix **??** for the experimental setup. In doing so, we intend to provide a review of the state-of-the-art on the variants of IRLS.

**Fixed Smoothing Parameter.** Instead of updating $\epsilon^{(t)}$ at each iteration, most works on IRLS use a fixed and small positive number, e.g., $\epsilon := \epsilon^{(t)} = 0.001$ for each $t$ [44, 76, 45, 47, 49, 51, 50, 77]. The intuition is to avoid division by the potentially very small residual $|a_i^\top x^{(t)} - y_i|$. But this common practice of fixing $\epsilon$ comes with at least three issues. First, IRLS with a fixed $\epsilon > 0$ converges only to an "$\epsilon$-approximate" point, not exactly the global minimizer ([78, 79, 44, 76], Figure 1a). Second, even if setting $\epsilon$ small (e.g., $\epsilon = 10^{-15}$) could lead to an accurate enough solution for $p = 1$ (Figure 1a), it can fail for $p < 1$ (Figure 1b). Finally, it makes obtaining a global linear convergence rate guarantee difficult: We are not aware of any theory about a global linear rate of IRLS with fixed $\epsilon$.

**Dynamic Smoothing Parameter.** Researchers have used different insights to reach the consensus of dynamically updating $\epsilon^{(t)}$ [26, 17, 80, 27, 81]. The first insight is that convergence to global minimizers ensues if $\epsilon^{(t)}$ is suitably decreased to 0 [26]. With $\beta \in (0, 1)$, one such decreasing rule is

$$\epsilon^{(t+1)} \leftarrow \beta\epsilon^{(t)} \text{ if certain conditions are satisfied, or keep } \epsilon^{(t+1)} \leftarrow \epsilon^{(t)} \text{ otherwise [17, 81].} \tag{5}$$

Figure 1a depicts the performance of [17] with an arbitrary choice $\beta = 0.5$ and $\epsilon^{(0)} = 1$. Also shown in Figure 1a are the IRLS methods with update rules of [26] and [27], which we discuss next.

The basis pursuit paper [26] also proposed a dynamic updating rule for $\epsilon^{(t)}$. In our context (1), it sets

$$\epsilon^{(0)} \leftarrow \infty, \quad \epsilon^{(t+1)} \leftarrow \min\left\{\epsilon^{(t)}, [r^{(t+1)}]_{\alpha+1}/m\right\}, \qquad r^{(t+1)} := Ax^{(t+1)} - y \in \mathbb{R}^m, \tag{6}$$

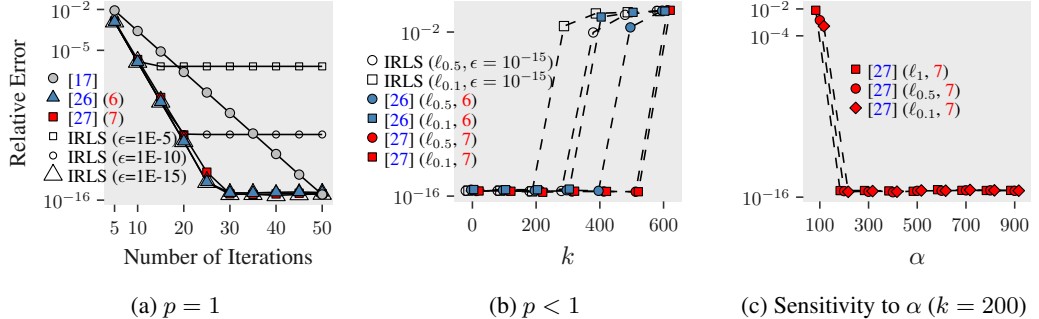

| (a) $p = 1$ | (b) $p < 1$ | (c) Sensitivity to $\alpha$ ($k = 200$) |

Figure 1: Figure 1a: The relative error $\|\boldsymbol{x}^{(t)} - \boldsymbol{x}^*\|_2 / \|\boldsymbol{x}^*\|_2$ at each iteration $t$ of different IRLS variants for $\ell_1$-regression ($k = 200$). Figure 1b: The relative error of $\texttt{IRLS}_p$ Algorithm 1 and that of [26] for $\ell_p$-regression (50 iterations). Figure 1c: The sensitivity of (6) and (7) to mis-specification of $\alpha$ (50 iterations). In Figure 1, we set $m = 1000, n = 10$, and results are averaged over 20 trials.

where the hyper-parameter[1] $\alpha$ is a non-negative integer, and $[\boldsymbol{r}^{(t+1)}]_{\alpha+1}$ is the $(\alpha + 1)$-th largest element of the residual $\boldsymbol{r}^{(t+1)}$ in absolute values. Clearly, (6) creates a non-increasing sequence of smoothing parameters $\epsilon^{(t)}$, and the decay rate of $\epsilon^{(t)}$ is adaptive to the data (in this case to the residual $\boldsymbol{r}^{(t+1)}$). While at first glance it is not clear how fast $\epsilon^{(t)}$ decays, [26] showed that the decay rate of $\epsilon^{(t)}$ in (6) is locally linear for $p = 1, \alpha = k$ under mild conditions, in accordance with the decay of the objective values.

The final dynamic update rule for the parameter $\epsilon^{(t)}$ is from [27][1] and improves upon (6) [26] via:

$$
\epsilon^{(0)} \leftarrow \infty, \qquad \epsilon^{(t+1)} \leftarrow \min\left\{\epsilon^{(t)}, \sigma^{(t+1)}/m\right\}
$$
$$
\text{where } \sigma^{(t+1)} \leftarrow \min\left\{\left\|\boldsymbol{r}^{(t+1)} - \boldsymbol{z}\right\|_1 : \boldsymbol{z} \in \mathbb{R}^m \text{ is } \alpha\text{-sparse}\right\}
\tag{7}
$$

Observe that (7) computes the $\ell_1$-norm $\sigma^{(t+1)}$ of the best $\alpha$-term approximation of the residual $\boldsymbol{r}^{(t+1)}$, which is in general larger than $[\boldsymbol{r}^{(t+1)}]_{\alpha+1}$ of (6), while both update rules (6) and (7) rely on the hyper-parameter $\alpha$. Ideally, if $\alpha = k$, and if IRLS with (6) or (7) converges to $\boldsymbol{x}^*$, then both $[\boldsymbol{r}^{(t+1)}]_{\alpha+1}$ and $\sigma^{(t+1)}$ will approach 0 for large enough $t$. While update rules (6) and (7) perform similarly for $\ell_1$-regression (Figure 1a), (6) fails more easily for small $p$ than (7) (Figure 1b). Finally, we note that overestimating the sparsity level $k$ by setting $\alpha > k$ deteriorates the performance of (7) only slightly (Figure 1c).

**Summary.** In Section 2.2 we delivered two take-away messages: (1) IRLS with a fixed $\epsilon^{(t)}$ finds some approximate solution, sometimes good enough, (2) dynamically updating $\epsilon^{(t)}$ as per (5)-(7) leads to local [26] or global [17, 27] linear convergence guarantees. In view of Figure 1, we will next consider the $\texttt{IRLS}_p$ Algorithm 1 with update rule (7) for $\epsilon^{(t)}$, and analyze its convergence rates.

## 3 Convergence Theory of IRLS for Robust Regression

In Section 3.1 we introduce the stable range space property and justify it as our core assumption. Under this assumption, we prove the global linear convergence (Theorem 1) and local superlinear convergence (Theorem 2) of the $\texttt{IRLS}_p$ Algorithm 1 in Sections 3.2 and 3.3, respectively.

### 3.1 The Stable Range Space Property

**Definition 1** (Stable Range Space Property)**.** A matrix $\boldsymbol{A} \in \mathbb{R}^{m \times n}$ is said to satisfy the *range space property (RSP)* of order $k$, or the $k$-RSP for short, if the following holds for any vector $\boldsymbol{d}$ in the range

---

[1]In [26] and [27], $\alpha$ is the sparsity level $k$, but $k$ might be unknown in practice, so we treat it as a hyper-parameter. That being said, in the experiments we set $\alpha = k$ by default, for simplicity and in light of Figure 1c.

space of $\boldsymbol{A}$ and any set $S \subset \{1, \ldots, m\}$ of cardinality at most $k$:

$$\sum_{i \in S} |d_i| < \sum_{i \in S^c} |d_i| \tag{8}$$

The *stable RSP* of $\boldsymbol{A}$ is defined in the same way as the RSP except that we now require

$$\sum_{i \in S} |d_i| \leq \eta \sum_{i \in S^c} |d_i| \tag{9}$$

for some $\eta \in (0, 1)$. We write "$(k, \eta)$-stable RSP" to emphasize the parameters $k$ and $\eta$.

By definition, the stable RSP implies the RSP. Note that the $(k, \eta)$-stable RSP was proposed in [53] to analyze a reweighted $\ell_1$-minimization algorithm for compressed sensing. With the notation of Proposition 1, we can see that checking (8) for all $\boldsymbol{d}$ in the range space of $\boldsymbol{A}$ is equivalent to checking it for all $\boldsymbol{d}$ in the nullspace of $\boldsymbol{D}$, the latter being the well-known *nullspace property* (NSP) [24, 28]. In other words, the (stable) RSP of $\boldsymbol{A}$ is equivalent to the (stable) NSP of $\boldsymbol{D}$.

In Sections 3.2 and 3.3, we will use the $(k, \eta)$-stable RSP as an assumption for analysis. Arguably, this is a weak assumption to make as the (stable) RSP is very close to a sufficient and necessary condition for exact recovery of the coefficient vector via $\ell_p$-robust regression (1):

**Proposition 2** (Exact Recovery $\Leftrightarrow$ RSP). *Let $p \in (0, 1]$. For all $\boldsymbol{x}^*$ and $\boldsymbol{y}$ such that $\boldsymbol{A}\boldsymbol{x}^* - \boldsymbol{y}$ is $k$-sparse $\boldsymbol{x}^*$ is the unique solution to (1) if and only if $\boldsymbol{A} \in \mathbb{R}^{m \times n}$ satisfies the $k$-RSP.*

It follows from [82, Theorem 5] and Proposition 1 that checking whether a given matrix $\boldsymbol{A}$ satisfies the stable RSP is *co-NP-complete* [83]. On the other hand, we show that random Gaussian matrices of size $m \times n$ with sufficiently large $m$ satisfy the stable RSP with high probability, which further justifies the usage of the stable RSP as an assumption in our analysis of $\texttt{IRLS}_p$:

**Proposition 3** (Gaussian $\Rightarrow$ RSP). *Suppose $m - n \geq 2k$. Let $\boldsymbol{A} \in \mathbb{R}^{m \times n}$ be a matrix with i.i.d. $\mathcal{N}(0, 1)$ entries. Let $\delta \in (0, 1)$ and $\eta \in (0, 1]$ be fixed constants. If it holds that*

$$\frac{(m - n)^2}{m - n + 1} \geq 2k \ln(em/k) \cdot \left( 1.67 + \eta^{-1} + \frac{\sqrt{18 \ln(2.5\delta^{-1})}}{\sqrt{2k \ln(em/k)}} \right)^2, \tag{10}$$

*then $\boldsymbol{A}$ satisfies the $(k, \eta)$-stable RSP with probability at least $1 - \delta$.*

The assumption $m - n \geq 2k$ of Proposition 3 is necessary, as it is not hard to prove that, if $m - n < 2k$ and $\boldsymbol{A}\boldsymbol{x}^* - \boldsymbol{y}$ is $k$-sparse, then the $\ell_0$-minimization problem $\min_{x \in \mathbb{R}^n} \|\boldsymbol{A}\boldsymbol{x} - \boldsymbol{y}\|_0$ has multiple solutions. With this assumption, (10) roughly becomes $m - n \geq ck \log(em/k)$ for some constant $c$ if $m - n$ is large enough; thus, ignoring logarithmic and constant factors, condition (10) becomes $m - n \geq \Theta(k)$, which is nearly optimal, as the condition $m - n \geq 2k$ is necessary.

Proposition 2 follows directly from [28, Theorem 4.9], so we omit its proof. The reader might also find Proposition 3 corresponds to [28, Corollary 9.34]. This corollary assumes $\boldsymbol{D}$ of (4) has i.i.d. $\mathcal{N}(0, 1)$ entries, from which is not immediate what the distribution of $\boldsymbol{A}$ is, thus it does not imply Proposition 3 directly; this is why we provide a complete proof for Proposition 3 in Appendix **??**.

## 3.2 Global Linear Convergence of $\texttt{IRLS}_1$

With the background provided in Section 2, we are now ready to state the first main result:

**Theorem 1** (Global Linear Convergence). *Suppose $\boldsymbol{A} \in \mathbb{R}^{m \times n}$ obeys the $(k, \eta)$-stable RSP with $\eta \in (0, 3/4)$ (cf. Definition 1). Let $\boldsymbol{A}\boldsymbol{x}^* - \boldsymbol{y}$ be $k$-sparse. If the smoothing parameter $\epsilon^{(t)}$ is updated as per (7) with $\alpha = k$ during the execution of the $\texttt{IRLS}_1$ Algorithm 1, then it holds for any $t \geq 1$ that*

$$\left\| \boldsymbol{A}\boldsymbol{x}^{(t+1)} - \boldsymbol{y} \right\|_1 - \left\| \boldsymbol{A}\boldsymbol{x}^* - \boldsymbol{y} \right\|_1 \leq \left( 1 - \frac{(3 - 4\eta)^2}{294\eta m} \right)^t \cdot 3 \cdot \left\| \boldsymbol{A}\boldsymbol{x}^{(1)} - \boldsymbol{y} \right\|_1, \tag{11}$$

*meaning that the $\texttt{IRLS}_1$ Algorithm 1 converges linearly and globally in objective value.*

**Discussion.** The condition $\eta \in (0, 3/4)$ is better understood via Proposition 3, which asserts that this condition holds with high probability if $\boldsymbol{A}$ has i.i.d. Gaussian entries and if $m$ is large enough

(compared to $n, k$). Moreover, if the $(k, \eta)$-stable RSP holds, Proposition 2 implies that $\boldsymbol{x}^*$ is a global minimizer of (1) with $p = 1$. The strength of Theorem 1 is in that it guarantees linear convergence for *any initialization, starting at the first iteration*. However, the price to pay is that the convergence rate seems conservative with constant $1/294$, and depends on the number $m$ of samples: (11) predicts that $\texttt{IRLS}_1$ converges with accuracy $\delta$ in $O(m \log(1/\delta))$ iterations and $m$ can be large. We argue that such dependency is not an artifact of our analysis, because similar dependencies have been established for IRLS variants for $\ell_p$-regression (1) with $p > 2$ [2, Theorem 3.1], for interior point methods in convex optimization with $m$ inequality constraints ([84, Chapter 11], [55, Chapter 5]), and for SGD in smooth & strongly convex optimization with a sum of $m$ function components [85, Theorem 2.1]). Moreover, an adversarial initialization for which the rate empirically depends on $m$ was pointed out for IRLS for basis pursuit in [27]. That being said, we conjecture that the linear dependency on $m$ in (11) can be improved to $\sqrt{m} \log m$ for some different choice of $\epsilon^{(t)}$; see [2] and [84, Sections 11.5 and 11.8.2] for why this conjecture makes sense. Furthermore, Figure 1a empirically depicts that, with the least-squares initialization, i.e., with $w_i^{(0)} = 1$ for all $i$, $\texttt{IRLS}_1$ typically converges to $\boldsymbol{x}^*$ in 30 iterations. This hints to the possibility of alleviating the dependency on $m$ by analyzing the least-squares initialization, a challenging task which we leave to future work.

**Comparison to [17].** To our knowledge, [17] is the only paper that claimed the global linear convergence of an IRLS variant with $\epsilon^{(t)}$ updated as per (5) and with $p = 1$. In particular, [17] sets $\epsilon^{(t+1)} \leftarrow \beta \epsilon^{(t)}$ whenever $\|\boldsymbol{x}^{(t+1)} - \boldsymbol{x}^{(t)}\|_2 \leq 2\beta \epsilon^{(t)}$; see Figure 1a for the convergence of IRLS under this rule. The update rule $\epsilon^{(t+1)} \leftarrow \beta \epsilon^{(t)}$ of [17] has the advantage of being faster to compute than (7). In spite of this, and of their beautiful proof idea, their result has several disadvantages compared to Theorem 1. First, their proof involves sophisticated probabilistic arguments, and is tailored towards a feature matrix that satisfies strong concentration properties. Our analysis, on the other hand, uses the $(k, \eta)$-stable RSP, which is close to a necessary and sufficient condition for the success of $\ell_1$-regression (cf. Proposition 2) and expected to hold for a much larger range of matrices than just (sub-)Gaussian matrices (see [86, 87] for related results in the context of basis pursuit). Second, some statements in their proof are inaccurate (e.g., their Lemma 5 does not hold for $k < n$, the last paragraph in their proving Lemma 8 is not rigorous). Third, their update rule incurs two hyper-parameters, $\epsilon^{(0)}$ and $\beta$, while Theorem 1 is based on the decreasing rule (7) that is adaptive to data, and the only hyper-parameter $\alpha$ is easier to set (Figure 1c).

**Connection to [27].** A related global linear convergence was established for the IRLS method for basis pursuit in [27, Theorem 3.2]. Our proof is inspired by [27], but also improves on its strategy. For example, their Theorem 3.2 involves two parameters for the *stable nullspace property*, while we only have a single RSP parameter $\eta$, simplifying matters. Also, we obtain a constant of $1/294$ in Theorem 1, which is better than the value of $1/768$ for the respective constant in [27, Theorem 3.2].

It is important to note that the proofs of [17, 27] and ours heavily rely on certain rules for decreasing the smoothing parameter $\epsilon^{(t)}$; all these proofs of global linear convergence for IRLS would break down if $\epsilon^{(t)}$ were fixed. While the main theorems of both [17] and [27] are limited to the $p = 1$ case, we present our Theorem 2 for $p \in (0, 1)$ next.

### 3.3 Local Superlinear Convergence of $\texttt{IRLS}_p$

The $\ell_p$-regression problem (1) with $p \in (0, 1)$ is more challenging than the case $p = 1$ due to the lack of convexity. However, here we show that at least locally, $\texttt{IRLS}_p$ converges with a superlinear rate of order $2 - p$. Moreover, since it is valid to run $\texttt{IRLS}_p$ even with $p = 0$, we carefully design the proof such that the following result holds not only for $p \in (0, 1]$, but also for $p = 0$, in which case $\texttt{IRLS}_p$ can be interpreted as an algorithm minimizing a sum-of-logarithm objective (see Appendix **??**).

**Theorem 2** (Local Superlinear Convergence). *Run $\texttt{IRLS}_p$ with $p \in [0, 1]$ and update $\epsilon^{(t)}$ by (7) with $\alpha = k$. Assume $\boldsymbol{A}$ satisfies the $(k, \eta)$-stable RSP (Definition 1). Let $c \in (0, 1)$ be a sufficiently small constant such that $2c^{1-p}\eta(\eta + 1) < (1 - c)^{2-p}$. Let $\boldsymbol{A}\boldsymbol{x}^* - \boldsymbol{y}$ be $k$-sparse with support $S^*$. Define*

$$\mu := 2\eta(\eta + 1)(1 - c)^{p-2} \cdot \min_{i \in S^*} |\boldsymbol{a}_i^\top \boldsymbol{x}^* - y_i|^{p-1}. \tag{12}$$

*If the initialization $\boldsymbol{x}^{(1)}$ is in a neighborhood of $\boldsymbol{x}^*$, in the sense that*

$$\left\| \boldsymbol{A}\boldsymbol{x}^{(1)} - \boldsymbol{A}\boldsymbol{x}^* \right\|_1 \leq c \cdot \min_{i \in S^*} |\boldsymbol{a}_i^\top \boldsymbol{x}^* - y_i|, \tag{13}$$

*then* $\mathrm{IRLS}_p$ *achieves the following superlinear convergence rate of order* $2 - p$ *for every* $t \geq 1$:

$$\left\| \boldsymbol{A}\boldsymbol{x}^{(t+1)} - \boldsymbol{A}\boldsymbol{x}^* \right\|_1 \leq \mu \cdot \left( \left\| \boldsymbol{A}\boldsymbol{x}^{(t)} - \boldsymbol{A}\boldsymbol{x}^* \right\|_1 \right)^{2-p} < \left\| \boldsymbol{A}\boldsymbol{x}^{(t)} - \boldsymbol{A}\boldsymbol{x}^* \right\|_1 \qquad (14)$$

*In particular, for* $p \in [0, 1)$*, the superlinear rate* (14) *implies*

$$\left\| \boldsymbol{A}\boldsymbol{x}^{(t+1)} - \boldsymbol{A}\boldsymbol{x}^* \right\|_1 \leq \left( \mu^{1/(1-p)} \cdot \left\| \boldsymbol{A}\boldsymbol{x}^{(1)} - \boldsymbol{A}\boldsymbol{x}^* \right\|_1 \right)^{(2-p)^t - 1} \cdot \left\| \boldsymbol{A}\boldsymbol{x}^{(1)} - \boldsymbol{A}\boldsymbol{x}^* \right\|_1. \qquad (15)$$

**Example 1** (*Quadratic versus linear rates*)**.** We illustrate the practical benefit of a superlinear (quadratic) convergence rate with the following simple calculation. Suppose that the inequality in (13) is barely fulfilled such that in the case of $p = 0$, $\boldsymbol{x}^{(1)}$ satisfies $\mu \cdot \|\boldsymbol{A}\boldsymbol{x}^{(1)} - \boldsymbol{A}\boldsymbol{x}^*\|_1 \leq 0.9999999$. Then (15) implies that after 30 iterations, the residual error is far below numerical precision already since $\|\boldsymbol{A}\boldsymbol{x}^{(31)} - \boldsymbol{A}\boldsymbol{x}^*\|_1 \leq 0.9999999^{2^{30}-1} \cdot \|\boldsymbol{A}\boldsymbol{x}^{(1)} - \boldsymbol{A}\boldsymbol{x}^*\|_1 \approx 10^{-47} \cdot \|\boldsymbol{A}\boldsymbol{x}^{(1)} - \boldsymbol{A}\boldsymbol{x}^*\|_1$. On the other hand, for $p = 1$, a linear convergence factor $\mu$ of $\mu \leq 0.9999999$ is only able to guarantee a decay of the order $\mu^{30} \leq 0.9999999^{30} \approx 0.999997$ after 30 iterations.

**Discussion.** Even though $\mathrm{IRLS}_p$ with $p \in (0, 1)$ is tailored to $\ell_p$-regression, we measure the progress of $\mathrm{IRLS}_p$ in $\ell_1$-norm, e.g., we provide upper bounds on $\|\boldsymbol{A}\boldsymbol{x}^{(t+1)} - \boldsymbol{A}\boldsymbol{x}^*\|_1$; this is because doing so avoids the use of Hölder's inequality, which allows us to give tighter results that improve on [26] (see the next paragraph). The second point that deserves discussion is the local convergence neighborhood defined by the right-hand side of (13). Note that, we typically assume that every outlier sample $(\boldsymbol{a}_i, y_i)$ (with $i \in S^*$) would result in a relatively large residual at $\boldsymbol{x}^*$, say $|\boldsymbol{a}_i^\top \boldsymbol{x}^* - y_i| \gg 0$. Hence the minimization term of the right-hand side in (13) is well-behaved. Next, one might ask how to obtain such an initialization $\boldsymbol{x}^{(1)}$ that satisfies (13). One might run $\mathrm{IRLS}_1$ to produce such an initialization; note though that the upper bound of (13) can not be computed, so one could not detect when to switch from $\mathrm{IRLS}_1$ to $\mathrm{IRLS}_p$ ($p \in [0, 1)$). Or alternatively, one could run $\mathrm{IRLS}_p$ with a least-squares initialization and count on empirical global (super)linear convergence of $\mathrm{IRLS}_p$; the latter is what we did in the experiments and is what we recommend. A final and important remark is that $\ell_p$-regression (1) with $p \in (0, 1)$ is in general NP-hard [28, Exercise 2.10], but this does not contradict the theoretical local superlinear convergence of Theorem 2 and the empirical global convergence in Figures 1b and 2b, and this does not mean that $\mathrm{IRLS}_p$ solves an NP-hard problem in polynomial time. The catch is that we operate under the assumption that $\boldsymbol{x}^*$ leads to a $k$-sparse residual and is a unique global minimizer of (1) (cf. Proposition 2). With this assumption and small enough $k/m$, $\ell_p$-regression is tractable and can be solved via $\mathrm{IRLS}_p$ to high accuracy (Figure 1b).

**Connection to [26].** [26, Theorem 7.9] proves the local superlinear convergence of IRLS with update rule (6) for $\ell_p$-basis pursuit (4), which motivates Theorem 2. However, [26, Theorem 7.9] requires $c$ of (13) to satisfy $c = O(1/m^{2/p-1})$, which means that, as $p \to 0$ or $m \to \infty$, the eligible value of $c$ quickly becomes vanishingly small, and thus the radius of local convergence (13) diminishes greatly. In contrast, in our condition $2c^{1-p}\eta(\eta + 1) < (1 - c)^{2-p}$, $c$ has no direct dependency on $m$ and is well-behaved even if $p \to 0$. The above difference is due partly to our improved proof strategy, and partly to the different update rules of the smoothing parameter $\epsilon^{(t)}$. Figure 1b showed that rule (6) of [26] does not work well for small $p$, in contrast to rule (7) that we use; a possible explanation for this phenomenon is that rule (6) decreases $\epsilon^{(t)}$ too fast, yielding a poor initialization for the next iteration of weighted least-squares (similarly to the interior point method, cf. [84]). Finally, [26, Theorem 7.9] does not hold for $p = 0$, for which Theorem 2 still holds and suggests a local quadratic rate.

## 4    Applications and Experiments

We now explore the performance of $\mathrm{IRLS}_p$ in three different applications, *real phase retrieval* (Section 4.1), *linear regression without correspondences* (Section 4.2), and *face restoration* (Section 4.3). Note that the first two applications are special examples of the recent algebraic-geometric framework called *homomorphic sensing* [88], [89, Section 4], [90, Section 1.2]. In Section 4.4, we examine the behavior of $\mathrm{IRLS}_p$ for different values of $p$ and under noise.

### 4.1    Real Phase Retrieval

In real phase retrieval [56, 57], we are given a measurement matrix $\boldsymbol{A} = [\boldsymbol{a}_1, \ldots, \boldsymbol{a}_m]^\top$ and $\boldsymbol{y} = [y_1, \ldots, y_m]^\top$, where $y_i := |\boldsymbol{a}_i^\top \boldsymbol{x}^*|$, and we need to find either $\boldsymbol{x}^*$ or $-\boldsymbol{x}^*$. This problem

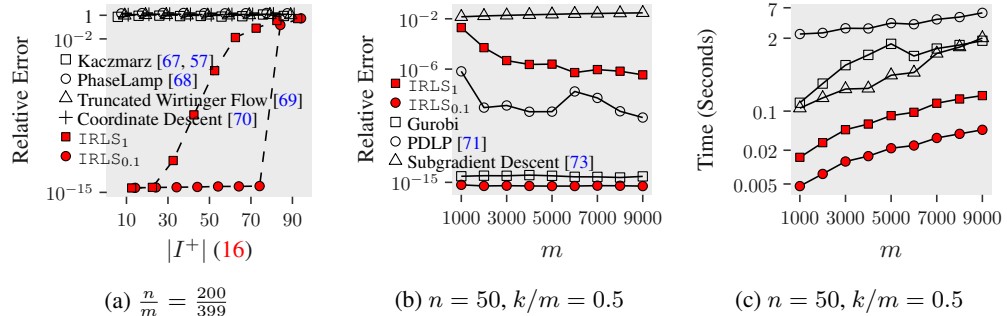

(a) $\frac{n}{m} = \frac{200}{399}$    (b) $n = 50, k/m = 0.5$    (c) $n = 50, k/m = 0.5$

Figure 2: Figure 2a: Relative error $\min\{\|\hat{\boldsymbol{x}} - \boldsymbol{x}^*\|_2, \|\hat{\boldsymbol{x}} + \boldsymbol{x}^*\|_2\}/\|\boldsymbol{x}^*\|_2$ of the methods that produce estimates $\hat{\boldsymbol{x}}$ for real phase retrieval. Figures 2b-2c: Relative errors $\|\hat{\boldsymbol{x}} - \boldsymbol{x}^*\|_2/\|\boldsymbol{x}^*\|_2$ and running times for linear regression without correspondences. $\texttt{IRLS}_p$ run for at most 50 iterations. 50 trials.

is a relative of the complex phase retrieval problem [65], where all data $\boldsymbol{y}$ and $\boldsymbol{A}$, as well as the ground-truth $\boldsymbol{x}^*$, are complex-valued, and which has applications in X-ray crystallography [91].

Here we show that real phase retrieval can be solved via $\ell_p$-regression (1). Consider the index sets

$$I^+ := \{i : \boldsymbol{a}_i^\top \boldsymbol{x}^* > 0\}, \qquad I^- := \{i : \boldsymbol{a}_i^\top \boldsymbol{x}^* < 0\}. \tag{16}$$

We might assume $\boldsymbol{a}_i^\top \boldsymbol{x}^* \neq 0$ for every $i$ without loss of generality. Then we know that either $\boldsymbol{A}\boldsymbol{x}^* - \boldsymbol{y}$ has its $\ell_0$ norm equal to $m - |I^-|$, or $\boldsymbol{A}(-\boldsymbol{x}^*) - \boldsymbol{y}$ has its $\ell_0$ norm equal to $m - |I^+|$. With such sparsity patterns on these two residuals, we can minimize (1), which serves as a (non-convex) relaxation of $\ell_0$-minimization, to recover $\boldsymbol{x}^*$ or $-\boldsymbol{x}^*$, whichever corresponds to a sparser residual. Here, we essentially treat one of the two clusters defined by (16) as inliers, and the other as outliers. This robust regression point of view on real phase retrieval seems to be known by experts [92, Section 1.2], but we have not found a paper that actually proposes to solve (1) for real phase retrieval.

Here we argue with Figure 2a that, solving (1) (by $\texttt{IRLS}_{0.1}$) for real phase retrieval can be beneficial. Indeed, theoretically, $\texttt{IRLS}_{0.1}$ converges to $\pm\boldsymbol{x}^*$ locally but superlinearly (as a corollary of Theorem 2). Empirically, Figure 2a shows that $\texttt{IRLS}_{0.1}$ succeeds in recovering $\pm\boldsymbol{x}^*$ by using only $m = 2n - 1$ samples, which is exactly the theoretical minimum for real phase retrieval [65, Proposition 2.5]. Figure 2a also shows that multiple other state-of-the-art methods[2] [57, 67, 68, 69, 70] fail to recover $\pm\boldsymbol{x}^*$ in this extreme situation, even though some of them have *nearly optimal* sample complexity, which requires roughly $O(n)$ samples up to a logarithmic factor to succeed (*caution*: nearly optimal $\neq$ optimal). As suggested by Proposition 3, for $|I^+|$ fixed, minimizing (1) with a Gaussian matrix $\boldsymbol{A}$ enjoys $m = O(n)$ sample complexity, up to also a logarithmic factor.

Despite these advantages, we also emphasize that solving (1) (via $\texttt{IRLS}_{0.1}$) for real phase retrieval has an inherent limit: The performance depends on the number $|I^+|$ of positive signs and the number $m$ of samples; we might call $|I^+|/m$ the inlier rate or outlier rate. At $n/m = 200/399$, we see that $\texttt{IRLS}_{0.1}$ succeeds if $|I^+| \leq 70$, and if $|I^+| \geq 399 - 70$ by symmetry, but it fails if $|I^+|$ and $|I^-|$ get closer (Figure 2a). For $\texttt{IRLS}_{0.1}$ to successfully handle the case $|I^+| = |I^-|$, we need more samples, empirically, say $m \geq 5n$. It is an interesting future direction to design an algorithm that can handle the case of minimum samples ($m = 2n - 1$) and balanced data ($|I^+| = |I^-|$).

## 4.2 Linear Regression without Correspondences

Compared to (real) phase retrieval, the problem of *linear regression without correspondences* is an also important, but less developed subject; see [58, 94, 59, 60, 61, 95, 96, 97] for recent advances. In this problem we are given $\boldsymbol{y} \in \mathbb{R}^m$ and $\boldsymbol{A} \in \mathbb{R}^{m \times n}$, and we need to solve the equations

$$\boldsymbol{y} = \boldsymbol{\Pi}\boldsymbol{A}\boldsymbol{x} \tag{17}$$

for some unknown permutation matrix $\boldsymbol{\Pi} \in \mathbb{R}^{m \times m}$ and vector $\boldsymbol{x} \in \mathbb{R}^n$, under the assumption that there exists a solution $(\boldsymbol{\Pi}^*, \boldsymbol{x}^*)$. Solving (17) is in general NP-hard for $n > 1$ [94, 98]. However, if

---

[2]Reviewing these algorithms is beyond the scope of this paper. Some of them are designed for complex phase retrieval but is applicable to the real case. We used the implementations from PhasePack [93].

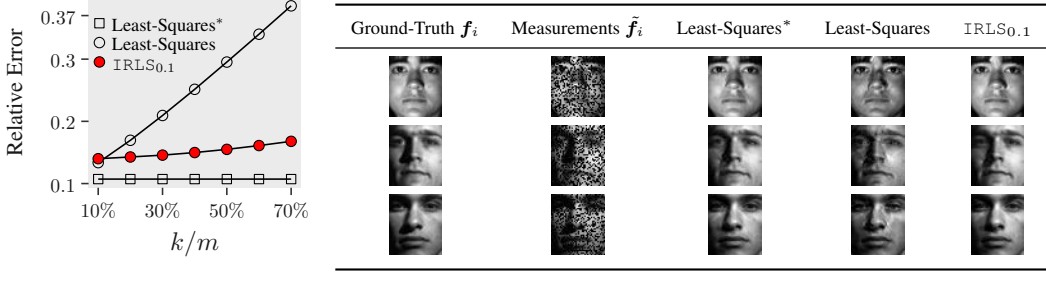

|  (a) Quantitative results | (b) Qualitative results (randomly selected images) |

Figure 3: Recover a face image of the Extended Yale B dataset [74] that is corrupted by *salt & pepper* noise [63]. Figure 3a: relative error as a function of the amount of random salt & pepper noise or sparsity level, averaged over 20 trials, all faces, and all individuals. Figure 3b: qualitative results.

we further assume that $\mathbf{\Pi}^*$ permutes at most $k$ rows of $\boldsymbol{A}$, then we know that the residual $\boldsymbol{A}\boldsymbol{x}^* - \boldsymbol{y}$ is $k$-sparse. This is an insight of [59], where it was proposed to estimate $\boldsymbol{x}^*$ by solving (1) with $p = 1$.

We show that $\texttt{IRLS}_p$ with $p \in (0, 1)$ is more suitable than several baselines for solving this robust regression problem reduced from linear regression without correspondences. As baselines, we use the PDLP [71] and Gurobi [72] solvers to solve (1) with $p = 1$ as a linear program, and we also use a subgradient descent method implemented in the FOM toolbox of Beck & Guttmann-Beck [73]. Figure 2b shows that $\texttt{IRLS}_{0.1}$ is the most accurate, and Figure 2c shows that $\texttt{IRLS}_{0.1}$ is 30 times faster than Gurobi and subgradient descent, and is more than 100 times faster than PDLP. Interestingly, $\texttt{IRLS}_1$ is not very competitive in terms accuracy (Figure 2b). Moreover, $\texttt{IRLS}_1$ is slower than $\texttt{IRLS}_{0.1}$, as $\texttt{IRLS}_{0.1}$ converges faster and thus terminates earlier than $\texttt{IRLS}_1$; we terminate $\texttt{IRLS}_p$ whenever the objective stops decreasing (up to a tolerance $10^{-15}$). Finally, see also [45, 48, 49, 50, 51, 52] where their IRLS variants outperform a different set of baselines (for different problems).

### 4.3 Face Restoration from Sparsely Corrupted Measurements

Here we consider a simple face restoration experiment on the Extended Yale B dataset [74], downsampled as per [99]. This dataset contains the face images of 38 individuals under about $n \approx 60$ different illuminations. Let $\boldsymbol{F} = [\boldsymbol{f}_1, \dots, \boldsymbol{f}_{n+1}]$ be the matrix of faces of the same individual, where each $\boldsymbol{f}_i \in \mathbb{R}^m$ is a (vectorized) face image with $m = 2,016$ pixels. We assume that $\boldsymbol{F}$ is approximately low-rank (which is true under the *Lambertian reflectance* assumption [100]), thus each $\boldsymbol{f}_i$ can be approximately represented as a linear combination of other faces of the individual, i.e., $\boldsymbol{f}_i \approx \boldsymbol{F}_i \boldsymbol{x}_i^*$ for some $\boldsymbol{x}_i^* \in \mathbb{R}^n$, where $\boldsymbol{F}_i \in \mathbb{R}^{m \times n}$ is the same as $\boldsymbol{F}$, except with the $i$-th column excluded.

In our experiments of Figure 3, we corrupt $\boldsymbol{f}_i$ by random salt & pepper noise [63] and obtain the noisy face $\tilde{\boldsymbol{f}}_i$ as our measurement; so $\boldsymbol{F}_i \boldsymbol{x}_i^* - \tilde{\boldsymbol{f}}_i$ is approximately sparse. We consider three methods to estimate $\boldsymbol{x}_i^*$ and thus to find an estimate $\hat{\boldsymbol{f}}_i$ of $\boldsymbol{f}_i$: (1) Least-Squares that solves (1) with $\boldsymbol{A} = \boldsymbol{F}_i$, $\boldsymbol{y} = \tilde{\boldsymbol{f}}_i$, and $p = 2$, (2) Least-Squares* that solves (1) with $\boldsymbol{A} = \boldsymbol{F}_i$, $\boldsymbol{y} = \boldsymbol{f}_i$, and $p = 2$, used as a golden baseline, and (3) $\texttt{IRLS}_{0.1}$ which solves (1) with $\boldsymbol{A} = \boldsymbol{F}_i$ and $\boldsymbol{y} = \tilde{\boldsymbol{f}}_i$. Figure 3a shows that $\texttt{IRLS}_{0.1}$ is more robust to salt & pepper noise, whose amount is $k/m$, than Least-Squares, and has lower error $\|\hat{\boldsymbol{f}}_i - \boldsymbol{f}_i\|_2 / \|\boldsymbol{f}_i\|_2$ (recall though that Least-Squares is statistically optimal for Gaussian noise). Figure 3b shows some images randomly chosen from the dataset; observe that Least-Squares tends to "average" the illumination in all images $\boldsymbol{F}_i$, while $\texttt{IRLS}_{0.1}$ delivers faithful restoration.

### 4.4 The Choice of $p$ and Performance of $\texttt{IRLS}_p$ Under Noise

Here we outline two experiments that illustrate two different aspects of $\texttt{IRLS}_p$ empirically.

In the first experiment, visualized in Figure 4a, we compare the decay of the relative errors of the iterates of $\texttt{IRLS}_p$ for different choices of $p$ in the case that the residual is exactly $k$-sparse. We observe that $\texttt{IRLS}_1$ exhibits a linear error decay on the one hand and a superlinear decay for $\texttt{IRLS}_p$ with $p = 0.5$ and $p = 0.1$ that accelerates as $p$ decreases towards 0, confirming the rates predicted by Theorem 1 and Theorem 2. ($p = 1, 0.5, 0.1$ are rather arbitrary choices in our experiments.)

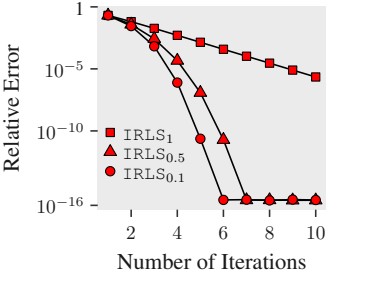
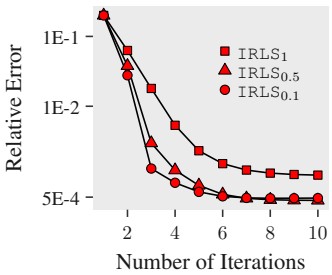

| (a) The exact $k$-sparse case | (b) The approximate $k$-sparse case |

Figure 4: Empirical convergence rates of $\texttt{IRLS}_p$ for different values of $p$ ($m = 1000, n = 10, k = 200$, averaged over 20 trials). Figure 4a: $\boldsymbol{Ax}^* - \boldsymbol{y}$ is exactly $k$-sparse. Figure 4b: $\boldsymbol{Ax}^* - \boldsymbol{y}$ is approximately $k$-sparse with $1\%$ noise (see Section **??** for data generation).

In another experiment, visualized in Figure 4b, we explore the robustness of $\texttt{IRLS}_p$ if the model assumption of $k$-sparse residuals is not satisfied, but only approximately. In particular, the input vector $\boldsymbol{y} \in \mathbb{R}^m$ of $\texttt{IRLS}_p$ is now such that its restriction on $S^*$ with $|S^*| = k$ has i.i.d. $\mathcal{N}(0,1)$ entries (corresponding to the sparse corruptions), but its other coordinates are distributed as independent Gaussian variables with mean $\boldsymbol{a}_i^\top \boldsymbol{x}^*$ and variance 0.01, where $i \in (S^*)^c$ (corresponding to dense noise); see also the code provided in Appendix **??**. We observe that also for this case of only approximately $k$-sparse residuals, $\texttt{IRLS}_p$ performs well, with faster and more accurate convergence for $p = 0.1$ and $p = 0.5$ compared to $p = 1$. We note that, strictly speaking, Theorem 1 and Theorem 2 do not apply directly to this case of approximately $k$-sparse residuals, but it is not hard to generalize it to this case; see [27, Theorem A.1] for a similar result for IRLS applied to basis pursuit.

## 5  Discussion and Future Work

In this work, we established novel global linear and local superlinear rates of $\texttt{IRLS}_p$ for $\ell_p$-regression (1) under the assumption of the stable range space property. Furthermore, we explored several applications for which $\texttt{IRLS}_p$ exhibits state-of-the-art results.

There are several directions that deserve further investigation. Theoretically, the Gaussian assumption of Proposition 3 might be weakened; similar results would hold for sub-Gaussian distributions [86, 87]. Note also that, in experiments (Figures 1 and 2), $\texttt{IRLS}_p$ works well even when $m - n \to 2k$. This leaves the question of whether the constant or logarithmic factors of condition (10) are too stringent for guaranteeing the stable RSP to hold; can this be improved? Finally, it is tempting to carry out a global rate analysis of $\texttt{IRLS}_p$ with $p \in (0,1)$ for $\ell_p$-regression; can one prove global (linear) rates under the stable RSP or other assumption?

Algorithmically, designing inexact solvers for the inner (weighted) least-squares problem (2) based on Krylov-subspace methods [101, 102] is expected to further accelerate our current IRLS implementation (our current implementation is attached in Appendix **??**). Also, our empirical experiments suggest that $\texttt{IRLS}_p$ (in particular $p \in (0,1)$) is worth being adopted and applied to many other outlier-robust estimation tasks beyond robust regression [45, 48, 49, 50, 51, 52, 103, 104, 105, 106, 107, 108].

### Acknowledgments and Disclosure of Funding

This work was supported by grants NSF 1704458, NSF 1934979, NSF-IIS-1837991, ONR MURI 503405-78051, and the NSF-Simons Collaboration on the Mathematical Foundations of Deep Learning (NSF grant 2031985).

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
