# A  IRLS as Smoothing Method Minimizing Quadratic Models

In this section, we clarify the precise relationship between the steps of $\texttt{IRLS}_p$ as described in Algorithm 1 and the $\ell_p$-objective on the residual, cf. (1).

## A.1  $\texttt{IRLS}_p$ as Lp-Regression for $0 < p \leq 1$

In Section 2.2, we shed light on the different rules for choosing the *smoothing parameter* $\epsilon^{(t+1)}$ in Algorithm 1. In fact, this terminology is used as there is an intimate relationship between the least squares step (2) of $\texttt{IRLS}_p$ and a *smoothed $\ell_p$-objective* $H_\epsilon(\cdot)$ which is, for given $\epsilon > 0$, defined as

$$H_\epsilon(\boldsymbol{r}) = \sum_{i=1}^{m} h_\epsilon(r_i) \tag{18}$$

where $h_\epsilon : \mathbb{R} \to \mathbb{R}$ is a symmetric smoothed one-dimensional function such that

$$h_\epsilon(r) = \begin{cases} \frac{1}{p}|r|^p & |r| > \epsilon \\ \frac{1}{2}\frac{r^2}{\epsilon^{2-p}} + \left(\frac{1}{p} - \frac{1}{2}\right)\epsilon^p & |r| \leq \epsilon \end{cases},$$

which makes $H_\epsilon$ a function akin to a scaled $\ell_p$-quasinorm, but which is quadratic around $0$. This definition corresponds to a scaled Huber loss in the case of $p = 1$ [10], but extends to the non-convex case if $0 < p < 1$.

By considering the derivative of $\epsilon \to h_\epsilon(r)$ for fixed $r$, it is easy to see that $h_\epsilon(r)$, and therefore also $H_\epsilon(\boldsymbol{r})$, is non-decreasing in $\epsilon$, which implies that

$$\frac{1}{p}\big\|\boldsymbol{Ax} - \boldsymbol{y}\big\|_p^p = \lim_{\epsilon' \to 0} H_{\epsilon'}(\boldsymbol{Ax} - \boldsymbol{y}) \leq H_\epsilon(\boldsymbol{Ax} - \boldsymbol{y}) \tag{19}$$

for any $\epsilon > 0$. We note that $H_\epsilon(\cdot)$ is continuously differentiable, so that the function $\mathbb{R}^m \times \mathbb{R}_{>0} \to \mathbb{R}, (\boldsymbol{r}, \epsilon) \mapsto H_\epsilon(\boldsymbol{r})$ can be considered as a *smoothing function* as defined in the non-smooth optimization literature [109, 110].

In this context, one interpretation of $\texttt{IRLS}_p$ (and of other IRLS methods such as [111, 26, 17, 27], with possibly different smoothing functions) is to consider it as a *smoothing method for $\ell_p$-regression with quadratic majorizing models*. Indeed, defining the function $Q_\epsilon : \mathbb{R}^m \times \mathbb{R}^m \to \mathbb{R}$,

$$Q_\epsilon(\boldsymbol{v}, \boldsymbol{r}) = \sum_{i=1}^{m} q_\epsilon(v_i, r_i), \text{ where } q_\epsilon(v, r) = h_\epsilon(r) + \frac{1}{2} \cdot \frac{v^2 - r^2}{\max\{|r|, \epsilon\}^{2-p}},$$

which is quadratic in $\boldsymbol{r}$, we observe that the least squares step (2) of $\texttt{IRLS}_p$ satisfies

$$\boldsymbol{x}^{(t+1)} = \operatorname*{argmin}_{\boldsymbol{x} \in \mathbb{R}^n} Q_\epsilon(\boldsymbol{Ax} - \boldsymbol{y}, \boldsymbol{r}^{(t)}) \tag{20}$$

where $\boldsymbol{r}^{(t)} = \boldsymbol{Ax}^{(t)} - \boldsymbol{y}$. Since $Q_\epsilon(\cdot, \boldsymbol{r})$ locally coincides with the smoothed function $H_\epsilon(\cdot)$ so that

$$H_\epsilon(\boldsymbol{r}) = Q_\epsilon(\boldsymbol{r}, \boldsymbol{r})$$

for each $\boldsymbol{r} \in \mathbb{R}^m$, and furthermore, since it is not hard to show that $Q_\epsilon(\cdot, \boldsymbol{r})$ majorizes $H_\epsilon(\cdot)$ in the sense that

$$H_\epsilon(\boldsymbol{v}) \leq Q_\epsilon(\boldsymbol{v}, \boldsymbol{r}) \tag{21}$$

for any $\boldsymbol{v}, \boldsymbol{r} \in \mathbb{R}^m$ and $\epsilon > 0$, it is possible to establish the monotonous decrease of the smoothed $\ell_p$-objective at each iteration of $\texttt{IRLS}_p$ without too much additional work: For each $t = 1, 2, \dots$ we obtain that

$$H_{\epsilon^{(t+1)}}\big(\boldsymbol{r}^{(t+1)}\big) \leq H_{\epsilon^{(t)}}\big(\boldsymbol{r}^{(t+1)}\big) \leq Q_{\epsilon^{(t)}}\big(\boldsymbol{r}^{(t+1)}, \boldsymbol{r}^{(t)}\big) \leq Q_{\epsilon^{(t)}}\big(\boldsymbol{r}^{(t)}, \boldsymbol{r}^{(t)}\big) = H_{\epsilon^{(t)}}\big(\boldsymbol{r}^{(t)}\big), \tag{22}$$

where the first inequality holds as long as $\epsilon^{(t+1)} \leq \epsilon^{(t)}$, which is satisfied for the smoothing parameter update rules described in Section 2.2.

The monotonicity outlined in (22) enables the following interpretation of the iterates $\{\boldsymbol{x}^{(t)}\}_{t \geq 1}$ of $\texttt{IRLS}_p$: Recalling the definition $\boldsymbol{r}^{(t)} = \boldsymbol{Ax}^{(t)} - \boldsymbol{y}$, it follows that without any assumption—in

particular, without assuming any range space property on the feature matrix $\boldsymbol{A}$—each accumulation point of $\{\boldsymbol{r}^{(t)}\}_{t\geq 1}$ is a first-order stationary point of the $\bar{\epsilon}$-smoothed $\ell_p$-objective $H_{\bar{\epsilon}}(\cdot)$ of (18) if $\bar{\epsilon}$ is defined as $\bar{\epsilon} = \lim_{t\to\infty} \epsilon^{(t)}$ (see, e.g., [26, Theorem 5.3] for a similar result).

We conclude this section by noting that $\texttt{IRLS}_p$ can be also interpreted within the framework of *majorization-minimization (MM)* [112, 113] algorithms—albeit, with the crucial difference that unlike for MM methods, the smoothing updates of $\texttt{IRLS}_p$ as described in Algorithm 1 change the underlying objective $H_{\epsilon^{(t)}}(\cdot)$ every time when $\epsilon^{(t)}$ changes, i.e., in many cases at each iteration.

## A.2 $\texttt{IRLS}_p$ as Sum-of-Logarithm Minimization for $p = 0$

The considerations of Section A.1 are tailored for $\texttt{IRLS}_p$ with $0 < p \leq 1$. However, it is valid to use $\texttt{IRLS}_p$ with $p = 0$ by setting the weights $w_i^{(t+1)}$ accordingly in (3). In this case, it is still possible to interpret $\texttt{IRLS}_0$ as a smoothing method with respect to an underlying smoothed surrogate objective.

In particular, we then define $H_\epsilon : \mathbb{R}^m \to \mathbb{R}$ as in (18) with

$$h_\epsilon(r) = \begin{cases} \log(|r|), & \text{if } |r| > \epsilon, \\ \frac{1}{2}\frac{r^2}{\epsilon^2} + \log(\epsilon) - \frac{1}{2}, & \text{if } |r| \leq \epsilon, \end{cases}$$

i.e., a smoothed *sum-of-logarithm* objective [114, 115, 116]. As the inequalities (21) and (22) still hold in this case, we conclude that $\texttt{IRLS}_0$ can be interpreted as a smoothing method for sum-of-logarithm minimization on the residuals $r = \boldsymbol{A}\boldsymbol{x} - \boldsymbol{y}$. We note that while there is a close relationship between their minimizers, a pointwise majorization of an $\ell_0$-objective $\|\boldsymbol{A}\boldsymbol{x} - \boldsymbol{y}\|_0$ by the smoothed objective $H_\epsilon(\cdot)$ as in (19) is *not* possible for $p = 0$.

Another minor difference to the case $0 < p \leq 1$ is that for $p = 0$, the smoothed objective $H_{\epsilon^{(t)}}(\boldsymbol{r}^{(t)})$ of $\texttt{IRLS}_0$ residuals $\boldsymbol{r}^{(t)}$ might converge to $-\infty$ as $\epsilon \to 0$; however, this is rather a technicality than a deterrence for the numerical performance of the algorithm, which remains excellent.

On the other hand, the local convergence analysis put forward in Section 3.3 and Theorem 2 still applies: In particular, we obtain that $\texttt{IRLS}_0$ as described in Algorithm 1 exhibits *local quadratic convergence* under appropriate assumptions on the feature matrix, cf. Theorem 2.

# B   Proofs of Main Results

## B.1   Global Linear Convergence for L1-Regression

*Proof of Theorem 1.* Define $\boldsymbol{r}^* := \boldsymbol{A}\boldsymbol{x}^* - \boldsymbol{y}$, $\boldsymbol{r}^{(t)} := \boldsymbol{A}\boldsymbol{x}^{(t)} - \boldsymbol{y}$, and $\boldsymbol{d}^{(t)} := \boldsymbol{r}^{(t)} - \boldsymbol{r}^* = \boldsymbol{A}\boldsymbol{x}^{(t)} - \boldsymbol{A}\boldsymbol{x}^*$. Recalling (22), for any $s > 0$ we have the following chain of inequalities:

$$H_{\epsilon^{(t+1)}}\left(\boldsymbol{r}^{(t+1)}\right) \leq H_{\epsilon^{(t)}}\left(\boldsymbol{r}^{(t+1)}\right) \leq Q_{\epsilon^{(t)}}\left(\boldsymbol{r}^{(t+1)}, \boldsymbol{r}^{(t)}\right) \leq Q_{\epsilon^{(t)}}\left(\boldsymbol{r}^{(t)} - s\boldsymbol{d}^{(t)}, \boldsymbol{r}^{(t)}\right) \quad (23)$$

In (23), the first inequality holds because $\epsilon^{(t+1)} \leq \epsilon^{(t)}$, and $\epsilon \mapsto H_\epsilon(\cdot)$ is a non-decreasing function in $\epsilon$, the second inequality holds because $H_\epsilon(\cdot)$ is majorized by $Q_\epsilon(\cdot, \times)$, see also (22). Furthermore, the third inequality holds due to the optimality of $\boldsymbol{r}^{(t+1)}$ or $\boldsymbol{x}^{(t+1)}$ (cf. (2) and (20)). From (23) we obtain

$$H_{\epsilon^{(t+1)}}\left(\boldsymbol{r}^{(t+1)}\right) - H_{\epsilon^{(t)}}\left(\boldsymbol{r}^{(t)}\right) \leq Q_{\epsilon^{(t)}}\left(\boldsymbol{r}^{(t)} - s\boldsymbol{d}^{(t)}, \boldsymbol{r}^{(t)}\right) - H_{\epsilon^{(t)}}\left(\boldsymbol{r}^{(t)}\right) \quad (24)$$

$$= \frac{1}{2}\sum_{i=1}^m \frac{\left(r_i^{(t)} - sd_i^{(t)}\right)^2 - \left(r_i^{(t)}\right)^2}{\max\{|r_i^{(t)}|, \epsilon^{(t)}\}} \quad (25)$$

$$= s\sum_{i=1}^m \frac{-r_i^{(t)} \cdot d_i^{(t)}}{\max\left\{|r_i^{(t)}|, \epsilon^{(t)}\right\}} + \frac{s^2}{2}\sum_{i=1}^m \frac{\left(d_i^{(t)}\right)^2}{\max\left\{|r_i^{(t)}|, \epsilon^{(t)}\right\}}. \quad (26)$$

We next derive upper bounds for the two sums of (26) respectively. Denote by $S^*$ the support of $\boldsymbol{r}^*$, and define $R := \{i : |r_i^{(t)}| > \epsilon\} \subset \{1, \ldots, m\}$. Then the first sum of (26) is

$$\sum_{i=1}^m \frac{-r_i^{(t)} \cdot d_i^{(t)}}{\max\{|r_i^{(t)}|, \epsilon^{(t)}\}} = \sum_{i \in S^*} \frac{-r_i^{(t)}}{\max\{|r_i^{(t)}|, \epsilon^{(t)}\}} \cdot d_i^{(t)} + \sum_{i \in (S^*)^c} \frac{-r_i^{(t)} \cdot r_i^{(t)}}{\max\{|r_i^{(t)}|, \epsilon^{(t)}\}} \tag{27}$$

$$\leq \sum_{i \in S^*} |d_i^{(t)}| - \sum_{i \in (S^*)^c \cap R} \frac{r_i^{(t)} \cdot r_i^{(t)}}{|r_i^{(t)}|} - \sum_{i \in (S^*)^c \cap R^c} \frac{r_i^{(t)} \cdot r_i^{(t)}}{\epsilon^{(t)}} \tag{28}$$

$$\leq (\eta - 1) \sum_{i \in (S^*)^c} |d_i^{(t)}| + \sum_{i \in (S^*)^c \cap R^c} |r_i^{(t)}| - \sum_{i \in (S^*)^c \cap R^c} \frac{r_i^{(t)} \cdot r_i^{(t)}}{\epsilon^{(t)}}. \tag{29}$$

In the last inequality, we applied $(k, \eta)$-RSP as our assumption, and used the fact $d_i^{(t)} = r_i^{(t)}$ for $i \in (S^*)^c$. Since $|r_i^{(t)}| \leq r_i^{(t)} \cdot r_i^{(t)}/\epsilon^{(t)} + \epsilon^{(t)}/4$, from the above we now arrive at

$$\sum_{i=1}^m \frac{-r_i^{(t)} \cdot d_i^{(t)}}{\max\{|r_i^{(t)}|, \epsilon^{(t)}\}} \leq (\eta - 1) \sum_{i \in (S^*)^c} |d_i^{(t)}| + |(S^*)^c \cap R^c| \cdot \frac{\epsilon^{(t)}}{4} \tag{30}$$

$$\leq (\eta - 1) \sum_{i \in (S^*)^c} |d_i^{(t)}| + \frac{m \cdot \epsilon^{(t)}}{4} \tag{31}$$

$$\leq (\eta - 1) \sum_{i \in (S^*)^c} |d_i^{(t)}| + \frac{1}{4} \sum_{i \in (S^*)^c} |d_i^{(t)}| = \left(\eta - \frac{3}{4}\right) \sum_{i \in (S^*)^c} |d_i^{(t)}|. \tag{32}$$

In the last inequality we used the definitions of $d_i^{(t)}$ and $\epsilon^{(t)}$. Next, the second summation of (26) is

$$\sum_{i=1}^m \frac{(d_i^{(t)})^2}{\max\{|r_i^{(t)}|, \epsilon^{(t)}\}} \leq \max_i |d_i^{(t)}| \cdot \frac{\sum_{i=1}^m |d_i^{(t)}|}{\epsilon^{(t)}} \tag{33}$$

$$\leq \eta(\eta + 1) \sum_{i \in (S^*)^c} |d_i^{(t)}| \cdot \frac{(\eta + 1) \sum_{i \in (S^*)^c} |d_i^{(t)}|}{\epsilon^{(t)}} \tag{34}$$

$$\leq \frac{49}{16} \eta \frac{\left(\sum_{i \in (S^*)^c} |d_i^{(t)}|\right)^2}{\epsilon^{(t)}}. \tag{35}$$

In the above we used $(k, \eta)$-stable RSP multiple times. With the two upper bounds on the sums of (26), and with $s' := s\left(\sum_{i \in (S^*)^c} |d_i^{(t)}|\right)$, we now obtain

$$H_{\epsilon^{(t+1)}}\left(\boldsymbol{r}^{(t+1)}\right) - H_{\epsilon^{(t)}}\left(\boldsymbol{r}^{(t)}\right) \leq s'\left(\eta - \frac{3}{4}\right) + \frac{(s')^2}{2} \cdot \frac{49}{16} \cdot \frac{\eta}{\epsilon^{(t)}}. \tag{36}$$

Since $s$ is an arbitrary non-negative integer, let $s$ be such that $s' = (12 - 16\eta)\epsilon^{(t)}/(49\eta)$. Then

$$H_{\epsilon^{(t+1)}}\left(\boldsymbol{r}^{(t+1)}\right) - H_{\epsilon^{(t)}}\left(\boldsymbol{r}^{(t)}\right) \leq -\frac{(3 - 4\eta)^2}{98\eta} \cdot \epsilon^{(t)}. \tag{37}$$

The definition of $\epsilon^{(t)}$ (7) implies there exists some $t' \leq t$ such that $\epsilon^{(t)} = \sigma^{(t')}/m$. Substituting this value of $s'$ into the above inequality gives us the bound

$$H_{\epsilon^{(t+1)}}\left(\boldsymbol{r}^{(t+1)}\right) - \|\boldsymbol{r}^*\|_1 \leq H_{\epsilon^{(t)}}\left(\boldsymbol{r}^{(t)}\right) - \|\boldsymbol{r}^*\|_1 - \frac{(3 - 4\eta)^2}{98\eta} \cdot \frac{\sigma^{(t')}}{m} \tag{38}$$

$$\leq H_{\epsilon^{(t)}}\left(\boldsymbol{r}^{(t)}\right) - \|\boldsymbol{r}^*\|_1 - \frac{(3 - 4\eta)^2}{98\eta} \cdot \frac{H_{\epsilon^{(t')}}\left(\boldsymbol{r}^{(t')}\right) - \|\boldsymbol{r}^*\|_1}{3m} \tag{39}$$

$$\leq H_{\epsilon^{(t)}}\left(\boldsymbol{r}^{(t)}\right) - \|\boldsymbol{r}^*\|_1 - \frac{(3 - 4\eta)^2}{98\eta} \cdot \frac{H_{\epsilon^{(t)}}\left(\boldsymbol{r}^{(t)}\right) - \|\boldsymbol{r}^*\|_1}{3m} \tag{40}$$

$$= \left(1 - \frac{(3 - 4\eta)^2}{294\eta m}\right) \cdot \left(H_{\epsilon^{(t)}}\left(\boldsymbol{r}^{(t)}\right) - \|\boldsymbol{r}^*\|_1\right). \tag{41}$$

The last two inequalities above are due to Lemma 1 and the monotonicity of $H_{\epsilon^{(t)}}(\boldsymbol{r}^{(t)})$ in $t$. Finally, Note that $H_{\epsilon^{(t+1)}}(\boldsymbol{r}^{(t+1)})$ majorizes $\|\boldsymbol{r}^{(t+1)}\|_1$ and Lemma 1 implies

$$H_{\epsilon^{(1)}}(\boldsymbol{r}^{(1)}) - \|\boldsymbol{r}^*\|_1 \leq 3\sigma^{(1)} \leq 3\|\boldsymbol{r}^{(1)}\|_1. \tag{42}$$

We have thus obtained

$$\|\boldsymbol{r}^{(t+1)}\|_1 - \|\boldsymbol{r}^*\|_1 \leq H_{\epsilon^{(t+1)}}(\boldsymbol{r}^{(t+1)}) - \|\boldsymbol{r}^*\|_1 \leq \left(1 - \frac{(3 - 4\eta)^2}{294\eta m}\right)^t \cdot 3\|\boldsymbol{r}^{(1)}\|_1. \tag{43}$$

and the proof is now complete. $\qquad\square$

## B.2 Local Superlinear Convergence for Lp-Regression

*Proof of Theorem 2.* We first show that the right inequality of (14), $\mu \cdot \|\boldsymbol{A}\boldsymbol{x}^{(t)} - \boldsymbol{A}\boldsymbol{x}^*\|_1^{1-p} < 1$, holds true. With (13) and the definition of $\mu$, this inequality holds whenever we have

$$2\eta(\eta + 1)(1 - c)^{p-2} \cdot \min_{i \in S^*} |\boldsymbol{a}_i^\top \boldsymbol{x}^* - y_i|^{p-1} \cdot \left(c \cdot \min_{i \in S^*} |\boldsymbol{a}_i^\top \boldsymbol{x}^* - y_i|\right)^{1-p} < 1, \tag{44}$$

$$\Leftrightarrow 2\eta(\eta + 1)(1 - c)^{p-2}c^{1-p} < 1, \tag{45}$$

which is true by the definition of $c$. This finishes proving the right inequality of (14). Next, one easily verifies (14) implies (15). Hence, it remains to prove the left inequality of (14).

Recall $\boldsymbol{r}^* := \boldsymbol{A}\boldsymbol{x}^* - \boldsymbol{y}$. Define $\boldsymbol{r}^{(t)} := \boldsymbol{A}\boldsymbol{x}^{(t)} - \boldsymbol{y}$, and $\boldsymbol{d}^{(t)} := \boldsymbol{r}^{(t)} - \boldsymbol{r}^* = \boldsymbol{A}\boldsymbol{x}^{(t)} - \boldsymbol{A}\boldsymbol{x}^*$. And recall that $S^*$ denotes the support of the ground-truth residual $\boldsymbol{r}^*$. We have

$$\sum_{i=1}^m \left(r_i^* + d_i^{(t+1)}\right)d_i^{(t+1)}w_i^{(t)} = 0, \tag{46}$$

a known property for the global minimizer $\boldsymbol{x}^{(t+1)}$ of the weighted least-squares problem (2). So

$$\sum_{i=1}^m \left(d_i^{(t+1)}\right)^2 w_i^{(t)} = -\sum_{i \in S^*} r_i^* d_i^{(t+1)} w_i^{(t)} \tag{47}$$

$$\leq \sum_{i \in S^*} |r_i^*| \cdot |d_i^{(t+1)}| \max\left\{|r_i^{(t)}|, \epsilon^{(t)}\right\}^{p-2} \tag{48}$$

$$\leq \sum_{i \in S^*} |r_i^*| \cdot |d_i^{(t+1)}| \cdot |r_i^{(t)}|^{p-2} \tag{49}$$

$$= \sum_{i \in S^*} |r_i^*| \cdot |d_i^{(t+1)}| \cdot |d_i^{(t)} + r_i^*|^{p-2} \tag{50}$$

$$\leq \sum_{i \in S^*} |r_i^*| \cdot |d_i^{(t+1)}| \cdot |r_i^*|^{p-2} \cdot (1 - c)^{p-2}. \tag{51}$$

In the last step, we used the fact $|d_i^{(t)}| \leq c \cdot |r_i^*|$ for all $i \in S^*$ (13), which means $|d_i^{(t)} + r_i^*| \geq (1 - c)|r_i^*|$. Thus, with the definition $C := (1 - c)^{p-2} \cdot \min_{i \in S^*} |\boldsymbol{a}_i^\top \boldsymbol{x}^* - y_i|^{p-1}$, we get

$$\sum_{i=1}^m \left(d_i^{(t+1)}\right)^2 w_i^{(t)} \leq \sum_{i \in S^*} |d_i^{(t+1)}| \cdot |r_i^*|^{p-1} \cdot (1 - c)^{p-2}$$
$$\leq C \sum_{i \in S^*} |d_i^{(t+1)}|. \tag{52}$$

From the above and the $(k, \eta)$-stable RSP, we can now arrive at

$$\sum_{i \in (S^*)^c} \left(d_i^{(t+1)}\right)^2 w_i^{(t)} \leq \sum_{i=1}^m \left(d_i^{(t+1)}\right)^2 w_i^{(t)} \leq C \sum_{i \in S^*} |d_i^{(t+1)}| \leq \eta \cdot C \sum_{i \in (S^*)^c} |d_i^{(t+1)}|. \tag{53}$$

Next, from the above and the Cauchy–Schwarz inequality, we get that

$$
\sum_{i\in(S^*)^c}|d_i^{(t+1)}| = \sum_{i\in(S^*)^c}|d_i^{(t+1)}|\cdot\big(w_i^{(t)}\big)^{\frac{1}{2}}\cdot\big(w_i^{(t)}\big)^{-\frac{1}{2}}
$$

$$
\leq \sqrt{\sum_{i\in(S^*)^c}\big(d_i^{(t+1)}\big)^2 w_i^{(t)}}\cdot\sqrt{\sum_{i\in(S^*)^c}\big(w_i^{(t)}\big)^{-1}} \tag{54}
$$

$$
\leq \sqrt{\eta\cdot C\sum_{i\in(S^*)^c}|d_i^{(t+1)}|}\cdot\sqrt{\sum_{i\in(S^*)^c}\big(w_i^{(t)}\big)^{-1}}.
$$

We next suppose $d_i^{(t+1)} = r_i^{(t+1)} - r_i^* \neq 0$ for some $i\in(S^*)^c$, for otherwise $\boldsymbol{r}^{(t+1)}$ is $k$-sparse and thus $\boldsymbol{x}^{(t+1)} = \boldsymbol{x}^*$ by the uniqueness of the $k$-sparse residual. Then the above inequality implies

$$
\sum_{i\in(S^*)^c}|d_i^{(t+1)}| \leq \eta\cdot C\sum_{i\in(S^*)^c}\big(w_i^{(t)}\big)^{-1}
$$

$$
= \eta\cdot C\cdot\sum_{i\in(S^*)^c}\max\big\{|r_i^{(t)}|,\epsilon^{(t)}\big\}^{2-p}
$$

$$
\leq \eta\cdot C\cdot\bigg(m\big(\epsilon^{(t)}\big)^{2-p} + \sum_{i\in(S^*)^c}|r_i^{(t)}|^{2-p}\bigg)
$$

$$
\leq \eta\cdot C\cdot\bigg(m\cdot\frac{\big(\sigma^{(t)}\big)^{2-p}}{m^{2-p}} + \sum_{i\in(S^*)^c}|d_i^{(t)}|^{2-p}\bigg) \tag{55}
$$

$$
\leq \eta\cdot C\cdot\bigg(\frac{1}{m^{1-p}}\cdot\Big(\sum_{i\in(S^*)^c}|d_i^{(t)}|\Big)^{2-p} + \Big(\sum_{i\in(S^*)^c}|d_i^{(t)}|\Big)^{2-p}\bigg)
$$

$$
\leq \eta\cdot C\cdot\Big(\frac{1}{m^{1-p}}+1\Big)\cdot\Big(\sum_{i=1}^{m}|d_i^{(t)}|\Big)^{2-p}
$$

$$
\leq 2\eta\cdot C\cdot\Big(\sum_{i=1}^{m}|d_i^{(t)}|\Big)^{2-p}.
$$

In the above, we used the definitions of $\epsilon^{(t)},\sigma^{(t)},r_i^{(t)},d_i^{(t)}$, and $S^*$. We have proved (14) since $\sum_{i=1}^{m}|d_i^{(t+1)}| \leq (1+\eta)\sum_{i\in(S^*)^c}|d_i^{(t+1)}|$ and by the definition of $C$. □

## C   Auxiliary Theoretical Results

**Lemma 1** (Upper Bound of The Residual). *Recall $\boldsymbol{r}^* = \boldsymbol{A}\boldsymbol{x}^* - \boldsymbol{y}$. Run* `IRLS`$_1$ *Algorithm 1 with $p=1$ and the update rule (7) for $\sigma^{(t)}$ and $\epsilon^{(t)}$, which yields the iterates $\{\boldsymbol{x}^{(t)}\}_{t>0}$. If A satisfies $(k,\eta)$-stable RSP, then for every $t\geq 1$ the residual $\boldsymbol{r}^{(t)} := \boldsymbol{A}\boldsymbol{x}^{(t)} - \boldsymbol{y}$ satisfies*

$$
H_{\epsilon^{(t)}}\big(\boldsymbol{r}^{(t)}\big) - \big\|\boldsymbol{r}^*\big\|_1 \leq 3\sigma^{(t)}, \tag{56}
$$

*where $H_{\epsilon^{(t)}}\big(\boldsymbol{r}^{(t)}\big)$ is the smoothed $\ell_p$-objective defined in (18) with $p=1$.*

*Proof.* Recall $R := \{i : |r_i^{(t)}| > \epsilon\} \subset \{1,\ldots,m\}$ and the definition of $H_{\epsilon^{(t)}}$ (18), we have

$$
H_{\epsilon^{(t)}}\big(\boldsymbol{r}^{(t)}\big) - \big\|\boldsymbol{r}^*\big\|_1 = \sum_{i\in R}r_i^{(t)} + \sum_{i\in R^c}\frac{1}{2}\Big(\frac{r_i^{(t)}}{\epsilon^{(t)}} + \epsilon^{(t)}\Big) - \big\|\boldsymbol{r}^*\big\|_1 \tag{57}
$$

$$
\leq \big\|\boldsymbol{r}^{(t)}\big\|_1 + |R^c|\cdot\epsilon^{(t)} - \big\|\boldsymbol{r}^*\big\|_1 \tag{58}
$$

$$
\leq \sigma^{(t)} + \big\|\boldsymbol{r}^{(t)}\big\|_1 - \big\|\boldsymbol{r}^*\big\|_1. \tag{59}
$$

Then it remains to prove $\left\|\boldsymbol{r}^{(t)}\right\|_1 - \left\|\boldsymbol{r}^*\right\|_1 \leq 2\sigma^{(t)}$. For this, define $S^{(t)}$ to be the support of the $k$ largest entries of $\boldsymbol{r}^{(t)}$ in absolute values, then $\sum_{i \in (S^{(t)})^c} \left|r_i^{(t)}\right| = \sigma^{(t)}$, and we have that

$$\sum_{i \in (S^{(t)})^c} \left|r_i^{(t)} - r_i^*\right| \leq \sum_{i \in (S^{(t)})^c} \left|r_i^{(t)}\right| + \sum_{i \in (S^{(t)})^c} \left|r_i^*\right| \tag{60}$$

$$= \sigma^{(t)} + \left\|\boldsymbol{r}^*\right\|_1 - \sum_{i \in S^{(t)}} \left|r_i^*\right| \tag{61}$$

$$= 2\sigma^{(t)} + \left\|\boldsymbol{r}^*\right\|_1 - \left\|\boldsymbol{r}^{(t)}\right\|_1 + \sum_{i \in S^{(t)}} \left|r_i^{(t)}\right| - \sum_{i \in S^{(t)}} \left|r_i^*\right| \tag{62}$$

$$\leq 2\sigma^{(t)} + \left\|\boldsymbol{r}^*\right\|_1 - \left\|\boldsymbol{r}^{(t)}\right\|_1 + \sum_{i \in S^{(t)}} \left|r_i^{(t)} - r_i^*\right|. \tag{63}$$

Since $\boldsymbol{A}$ satisfies $(k, \eta)$-stable RSP and $\boldsymbol{r}^{(t)} - \boldsymbol{r}^*$ is in the range space of $\boldsymbol{A}$, it holds that

$$\sum_{i \in S^{(t)}} \left|r_i^{(t)} - r_i^*\right| \leq \eta \sum_{i \in (S^{(t)})^c} \left|r_i^{(t)} - r_i^*\right| \tag{64}$$

$$\Rightarrow \sum_{i \in S^{(t)}} \left|r_i^{(t)} - r_i^*\right| \leq \frac{\eta}{1 - \eta} \left(2\sigma^{(t)} + \left\|\boldsymbol{r}^*\right\|_1 - \left\|\boldsymbol{r}^{(t)}\right\|_1\right). \tag{65}$$

In the last inequality we used (63) and rearranged terms. Using (63) and (65) we obtain

$$\left\|\boldsymbol{r}^{(t)}\right\|_1 - \left\|\boldsymbol{r}^*\right\|_1 \leq \left\|\boldsymbol{r}^{(t)} - \boldsymbol{r}^*\right\|_1 \tag{66}$$

$$= \sum_{i \in (S^{(t)})^c} \left|r_i^{(t)} - r_i^*\right| + \sum_{i \in S^{(t)}} \left|r_i^{(t)} - r_i^*\right| \tag{67}$$

$$\leq 2\sigma^{(t)} + \left\|\boldsymbol{r}^*\right\|_1 - \left\|\boldsymbol{r}^{(t)}\right\|_1 + 2 \sum_{i \in S^{(t)}} \left|r_i^{(t)} - r_i^*\right| \tag{68}$$

$$\leq \frac{1 + \eta}{1 - \eta} \left(2\sigma^{(t)} + \left\|\boldsymbol{r}^*\right\|_1 - \left\|\boldsymbol{r}^{(t)}\right\|_1\right) \tag{69}$$

which implies

$$\left\|\boldsymbol{r}^{(t)}\right\|_1 - \left\|\boldsymbol{r}^*\right\|_1 \leq \frac{1 + \eta}{2} \cdot 2\sigma^{(t)} \leq 2\sigma^{(t)}. \tag{70}$$

This finishes the proof. □

## C.1 The Stable Range Space Property of Gaussian Matrices

*Proof of Proposition 3.* We need the notion of *Gaussian width* $w(\cdot)$ of a given set $\mathcal{K}$, defined as

$$w(\mathcal{K}) := \mathbb{E}_{\boldsymbol{g} \sim \mathcal{N}(0, \boldsymbol{I}_m)} \left[\sup_{\boldsymbol{r} \in \mathcal{K}} \boldsymbol{g}^\top \boldsymbol{r}\right]. \tag{71}$$

Write $\mathbb{S}^{m-1} := \{v \in \mathbb{R}^m : v^\top v = 1\}$. Denote by $\mathrm{Gr}(n, m)$ the set of $n$ dimensional subspaces of $\mathbb{R}^m$, also known as the *Grassmannian* manifold or variety. With $\eta \in (0, 1]$, consider the set

$$\mathcal{T}_k := \left\{\boldsymbol{r} \in \mathbb{R}^m : \sum_{i \in S} |r_i| > \eta \sum_{i \in S^c} |r_i|, \text{ for some } S \subset \{1, \ldots, m\} \text{ with of cardinality } k\right\}. \tag{72}$$

The following lemma gives an upper bound on the Gaussian width of $\mathcal{T}_k \cap \mathbb{S}^{m-1}$:

**Lemma 2** (Remark 9.30 and Proposition 9.33 of [28]). *If $m \geq 2k$ then we have:*

$$w\left(\mathcal{T}_k \cap \mathbb{S}^{m-1}\right) \leq \sqrt{2k \ln(em/k)} \cdot \left(1.67 + \eta^{-1}\right). \tag{73}$$

Lemma 2 and (10) imply that $w\big(\mathcal{T}_k \cap \mathbb{S}^{m-1}\big) \leq \frac{m-n}{\sqrt{m-n+1}}$. Since $\boldsymbol{A}$ has i.i.d. $\mathcal{N}(0,1)$ entries, we can think of its range space $\mathrm{r}(\boldsymbol{A})$ as drawn uniformly at random from the Grassmannian $\mathrm{Gr}(n,m)$. Invoking Lemma 3 with $\mathcal{V} = \mathrm{r}(\boldsymbol{A})$ and $\mathcal{M} = \mathcal{T}_k \cap \mathbb{S}^{m-1}$, we see that

$$\Pr\big(\mathrm{r}(\boldsymbol{A}) \cap \mathcal{T}_k \cap \mathbb{S}^{m-1} = \varnothing\big) \geq 1 - 2.5\exp\left(-\frac{1}{18}\Big(\frac{m-n}{\sqrt{m-n+1}} - w(\mathcal{T}_k \cap \mathbb{S}^{m-1})\Big)^2\right). \quad (74)$$

Furthermore, Lemma 2 and condition (10) make sure that

$$\frac{m-n}{\sqrt{m-n+1}} - w(\mathcal{T}_k \cap \mathbb{S}^{m-1}) \geq \frac{m-n}{\sqrt{m-n+1}} - \sqrt{2k\ln(em/k)} \cdot \Big(1.69 + \eta^{-1}\Big) \quad (75)$$

$$\geq \sqrt{18\ln(2.5\delta^{-1})}. \quad (76)$$

Combining the above leads us to

$$\Pr\big(\mathrm{r}(\boldsymbol{A}) \cap \mathcal{T}_k \cap \mathbb{S}^{m-1} = \varnothing\big) \geq 1 - 2.5\exp\big(-\ln(2.5\delta^{-1})\big) = 1 - \delta. \quad (77)$$

The event $\mathrm{r}(\boldsymbol{A}) \cap \mathcal{T}_k \cap \mathbb{S}^{m-1} = \varnothing$ implies that the stable $(k,\eta)$-stable RSP holds true. □

**Lemma 3** (Gordon's Escape Through a Mesh Theorem, Corollary 3.4 of [117], Theorem 4.3 of [118]). *Let $\mathcal{V}$ be a $n$-dimensional subspace of $\mathbb{R}^m$ drawn uniformly at random from the Grassmannian $Gr(n,m)$. Let $\mathcal{M}$ be a subset[3] of $\mathbb{S}^{m-1}$. If $w(\mathcal{M}) < \frac{m-n}{\sqrt{m-n+1}}$ then*

$$Pr\big(\mathcal{V} \cap \mathcal{M} = \varnothing\big) \geq 1 - 2.5\exp\left(-\frac{1}{18}\Big(\frac{m-n}{\sqrt{m-n+1}} - w(\mathcal{M})\Big)^2\right). \quad (78)$$

## D Experimental Setup

### D.1 How We Run Other Methods

In the real phase retrieval experiment (Figure 2a), all other baselines are implemented in PhasePack [93]. We use the following (quite standard) parameters:

```
opts.tol = 1e-11; opts.initMethod = 'Truncatedspectral';
```

In the experiments of linear regression without correspondences (Figures 2b and 2c), the implementation of PDLP [71] is here, and we run it with the command:

```
julia --project=scripts scripts/solve_qp.jl \
--instance_path test/trivial_lp_model.mps --iteration_limit 500 \
--method pdhg --output_dir [my directory]
```

Note that we run PDLP for a maximum of 500 iterations, while the recommended number of maximum iterations from their GitHub repo is 5000; this is mainly for the sake of efficiency. We use the newest version 9.5.1 of the Gurobi solver with default parameters. we employ the implementation of the FOM toolbox [73] for the (proximal) subgradient descent method. In particular, we invoke the function `prox_subgradient` with the function $G$ being zero and $F$ being $\|\boldsymbol{Ax} - \boldsymbol{y}\|_1$. We use 0 as initialization, and we observe similar performance when using the least-squares initialization. We set `par.alpha` to be $1/\|\boldsymbol{A}\|_2$, which corresponds to a stepsize of $1/\|\boldsymbol{A}\|_2/(t+1)$, where $t$ is the number of iterations. We make this choice because the default `par.alpha = 1` does not work well and sometimes diverges. Finally, we set the maximum number of iterations to be 10000, as the default choice 1000 leads to an estimate with large errors.

### D.2 Synthetic Data Generation

In this section, we provide the Matlab codes generating synthetic data for experiments visualized in Figures 1, 2, and 4, detailing the precise data generation process.

---

[3]The proof of Gordon [117] assumes $\mathcal{M}$ to be closed, but this assumption can be removed in view of the definition of the Gaussian width and the compactness of $\mathbb{S}^{m-1}$. The factor 3.5 of [117] can be improved to 2.5 [118]. To correct [118]; their condition "$w(S) < \sqrt{k}$" should be replaced by "$w(S) < k/\sqrt{k+1}$".

```
%% robust regression
function [y, A, x] = gen_RR(m, n, k, sigma)
    A = randn(m, n); x = randn(n,1);

    y = zeros(m,1);

    idx = datasample(1:m, k, 'Replace', false);

    y(idx) = randn(k,1);

    s = setdiff(1:m, idx);
    y(s) =  A(s, :) * x + sigma * randn(length(s), 1);
end

%% linear regression without correspondences
function [y, A, x] = gen_SLR(m, n, sigma, shuffle_ratio)
    A = randn(m, n); x = randn(n,1);

    w = sigma*randn(m, 1);

    y0 = A*x;
    k = int64(shuffle_ratio * m);
    partial_idx = datasample(1:m, k, 'Replace', false);
    y1 = y0(partial_idx, 1);
    y0(partial_idx, 1) = y1(randperm(k));

    y = y0 + w;
end

%% real phase retrieval
function [y, A, x] = gen_RPR(m, n, num_positive_sign)
    A = randn(m, n); x = randn(n,1); y = zeros(m,1);

    idx = datasample(1:m, num_positive_sign, 'Replace', false);

    y(idx) = A(idx, :) * x;

    s = setdiff(1:m, idx);
    y(s) = - A(s, :) * x;

    % make y positive
    idx = y < 0;
    y(idx) = -y(idx);

    A(idx,:) = -A(idx,:);
end
```

## E   Implementation

The below is our Matlab code that implements $\mathrm{IRLS}_p$:

```
function [x_hat] = IRLSp(A, y, p, k, num_iter)
    [m, n] = size(A);

    q = 2 - p;   l = m - k; epsilon = inf;

    w = ones(m,1); x_old = zeros(n,1);
```

```
    for i = 1:num_iter
        x_hat = (A' * (w.*A)) \  (A' * (w.* y));

        abs_residual = abs(A*x_hat - y);

        sigma = sum(mink(abs_residual, l)) / m;

        epsilon = max(min(epsilon, sigma), 1e-16);

        w = 1./ (max(abs_residual, epsilon)).^(q);

        if norm(x_old - x_hat)/ norm(x_hat) < 1e-15
            break;
        end
        x_old = x_hat;
    end
end
```