# OpenReview forum: "Global Linear and Local Superlinear Convergence of IRLS for Non-Smooth Robust Regression"
_NeurIPS.cc/2022/Conference — NeurIPS 2022 Accept_

### Official Review · Reviewer_yi3J · 2022-07-11

**Rating:** 6
**Confidence:** 4
**Soundness:** 3 good
**Presentation:** 4 excellent
**Contribution:** 3 good

**Summary:**

This work analyzes the famous IRLS algorithm with any choice of $p\in [0,1]$ and the adaptively updated smoothing parameter $\epsilon_t$. It is able to establish global linear convergence theory for $p=1$ with arbitrary initialization. Unlike previous works, it also establishes local superlinear convergence theory with sufficiently good initialization, for any $p\in [0,1]$. Experiments confirm the advantages of IRLS with $p<1$.

**Questions:**

1. Would it be possible to analyze the radius of the convergence zone (RHS of eq. 13) under some probabilistic assumptions. For example, one may consider the entries of $A$, $x^*$ are i.i.d standard normal, $y = Ax^* +r^*$, where $r^*$ is k-sparse. The coordinates of nonzero elements of $r^*$ are randomly sampled, and its nonzero elements follow any smooth distribution with zero mean and finite second order moments. Would the RHS of (13) converge to 0 as $k=\lambda m$, $n,m\to \infty$ where $\lambda$ is a small constant? I suspect that it is true (since you take a min of a set of random variables that has a growing cardinality), but I would like the authors to confirm or disprove my conjecture.

2. Following my point 1, if my conjecture is true, would it contradict your claim in line 66 that your radius of local convergence does not diminish as $m\to \infty$? It probably won't diminish if $k$ is fixed even when $m\to \infty$ (not $k=\lambda m$), but the author should add some discussions about which assumption is more reasonable, and which one is more common in real practice. In some sense, $k$ should grow with $m$, but I welcome the author(s) to provide me with some counterexamples in real applications.

3. As for the experiments, the performance of IRLS-0.1 is very impressive. Is $p=0.1$ typically (approximately) the best choice in all experiments? What if you choose $p=0.05$? Also, how the IRLS-1 and IRLS-0.1 are sensitive to the choice of update rules of smoothing parameters (as you introduced in section 2.2)? Is the current update rule of $\epsilon$ critical to the success of the presented IRLS-0.1? It might be good to add some simple experiments in the supplementary material.




**Limitations:**

Yes.

**Strengths And Weaknesses:**

Strengths:

1. This work improved the analysis and the result of the previous global convergence theory for IRLS with $p=1$.

2. The authors are able to show local superlinear convergence for any $p\in [0,1]$, which is not done in previous works.

3. The paper is very well-written. The previous works are reviewed in detail. The contributions are highlighted. The content is well-structured.

Weaknesses:

1. Although the stable RSP condition is satisfied for Wigner matrices, it does not seem to be easy to verify in general (one has to check all subsets of coordinates of size k). I understand that it was also adopted in some previous works, but it would be nice to have a condition that can be more easily verified in a general setting.

2. The radius of the convergence zone (RHS of eq. 13) for the nonconvex IRLS depends on the min over the nonzero residuals. Thus, the radius could be small when k (the number of nonzero elements), n, m are all large (consider an extreme case where $k,n,m\to \infty$), unless one can show that the smallest nonzero element is sufficiently far away from zero.


This work has made a nice contribution to the theory of the iteratively reweighted least squares, although there is still room for improvement on the theory side.

---

> ### Author Response · Authors · 2022-08-02
> **Replies to Individual Comments (Reviewer yi3J)**
>
> > Although the stable RSP condition is satisfied for Wigner matrices, it does not seem to be easy to verify in general (one has to check all subsets of coordinates of size k). I understand that it was also adopted in some previous works, but it would be nice to have a condition that can be more easily verified in a general setting.
> >
> > > **Reply**: We totally agree that the stable RSP condition is not easy to verify. This impediment is an inherent drawback that our analysis shares with hundreds of papers in the compressed sensing and matrix recovery literature where the null space property (NSP) or restricted isometry property (RIP) is used. In fact, it has been shown that the RIP is NP-hard to verify, and this is likely to be true also for the NSP and RSP.
> > >
> > > We also agree that it is highly desirable to discover alternative and easy-to-verify conditions that guarantee exactly/stable recovery for related problems (compressed sensing, matrix recovery, etc.), yet this is really an open problem (to our knowledge). In fact, it is very difficult, if not impossible, to do so, as the problem under investigation itself is computationally hard. As a compromise, one might instead prove that a condition (be it RSP, NSP, or RIP) holds under certain probabilistic assumptions on data, as we did in Proposition 3. This compromise might be satisfactory (depending on the research tastes of individuals); after all, we are tackling NP-hard problems (Lines 230-235).
>
> > Would it be possible to analyze the radius of the convergence zone (RHS of eq. 13) under some probabilistic assumptions. ...... I suspect that it is true (since you take a min of a set of random variables that has a growing cardinality), but I would like the authors to confirm or disprove my conjecture.
> >
> > > **Reply**: That probabilistic reasoning is wonderful. You are entirely correct and we agree that the minimization term would be zero as k goes to infinity. And if that is the case, our local convergence radius would be vanishingly small! On the other hand, as we have explained in our "Replies to Common Concerns", we believe this minimization is a reasonable term, as it measures the "outlier-ness" of the data (see our replies above for details). This an important point that needs further justification in our revision, though.
>
> > How IRLS-1 and IRLS-0.1 are sensitive to the update rules of the smoothing parameters, and is the current update rule of the smoothing parameter critical to the success of IRLS-0.1? It might be good to add some simple experiments in the supplementary material.
> >
> > > **Reply**: Yes, the current update rule of the smoothing parameter is critical to the success of IRLS-0.1. To be more specific, let us recall the dynamic update rules  (5), (6), (7) respectively. In [15], rule (5) is only applied to the case p=1 (See Lines 183-196 for discussion about (5)). It was observed already in [24, Section 8.1] that IRLS-p using rule (6) does not exhibit a good global convergence behavior if $p < 0.5$ (for the compressed sensing problem), which is consistent to what we observe also for robust regression.
> > >
> > > The update rules (6) and (7) are further compared for Lp minimization in Figure 1b, and we see that (7) performs much better. From a theoretical point of view as well, the local superlinear convergence radius implied by rule (7) is much larger than the one implied by rule (6), as discussed in Lines 236-244.
> > >
> > > Note that the only difference between (6) and (7) is that (7) takes the best $\alpha$-term approximation, while (6) takes the ($\alpha+1$)-th largest element from the residual. Without a proof, we suspect that the reason that (6) performs worse is because the smoothing parameter of (6) is decreased too fast at each iteration, resulting in sub-optimality (consider a similar situation where the regularization parameter for the log-barrier function should not be increased too fast in the interior point method).
> > >
> > > As a summary, it can be said that both theory and empirical behavior of IRLS are sensitive to the precise choice of the update rule, and are the reason why we choose rule (7). While both rule (6) and (7) enjoy empirical global linear convergence for p=1 (Figure 1a), we are only able to prove this for (7). Finally, it should be remarked that, even though (7) is the best update rule that we are aware of and it is crucial for our analysis, it is unclear to us how to design an optimal update rule for the smoothing parameter that can further improve IRLS.
>
> > **Promised Revision**: In our revision, we will mention that the hardness of computationally verifying the RSP. Also, we will also discuss (mathematically) why update rule (7) is important to the proofs.

---

> > ### Comment · Reviewer_yi3J · 2022-08-08
> > **Response to the authors**
> >
> > I thank the authors for the detailed response, and it addressed all of my concerns. In my opinion it is a solid paper, and I prefer not to change my score.

---

### Official Review · Reviewer_GXHV · 2022-07-11

**Rating:** 6
**Confidence:** 4
**Soundness:** 3 good
**Presentation:** 2 fair
**Contribution:** 3 good

**Summary:**

In this paper, the authors propose a new variant of iteratively reweighted least square (IRLS) algorithm and studied its theoretical property as well as the practical performance. The authors stated the stable range space property, which plays the fundamental role in the theoretical analysis of the new IRLS method. The authors showed that for the case of $p = 1$, the IRLS can achieve a global linear convergence rate. The authors also presented that for the case of $p \in (0, 1)$, the IRLS reach a superlinear convergence rate of the order of $2 - p$. Both local and global convergence rates can match the state-of-art results. The authors also conduct numerical experiments on different applications. And the empirical results showed that the performance of the IRLS algorithm is competitive against other methods.

**Questions:**

One question for the structure of the paper.

Could the authors put the discussion and future work on page 9 of the paper into one independent conclusions section? This can make the paper more clean and clear.

**Limitations:**

The authors have presented the limitations of this paper in the final page of the submission.

**Strengths And Weaknesses:**

This paper has the following strengths in terms of the originality, significance, quality and clarity:

Originality and significance: as far as we know, this paper provides the one of the first novel IRLS algorithm that could reach linear rate for the global convergence and superlinear rate for the local convergence. This is the main and independent theoretical contribution of this paper.

Quality and clarity: the authors provides detailed proof in the theoretical section and the consistent empirical results in the numerical experiments section. These empirical results show that the performance of the proposed IRLS method is competitive against other state-of-art algorithms in the application areas of real phase retrieval, regression without correspondences and robust face restoration. The new IRLS algorithm is specially good at handling the outliers, which is the key property in the problem of robust $l_p$ quasinorm regression. The paper is also organized with clear and clean structure and the language is well-organized so that the readers can easily understand the content of this submission.

However, despite the above advantages, the paper has the following weaknesses.

 The theoretical results only hold under the assumptions that $m$ must be much larger than the $n$ and $k$. However, in the numerical experiments, the IRLS algorithm also performs well under the condition that $m$ is comparable larger than $k$ and $n$. Hence, there is some gap between the theoretical analysis and the numerical experiments. Also the authors should mention in the paper any theoretical results for the case of $m$ is not large enough, i.e.,  assumption (10) doesn't hold. Is there any significant results for this special case?

Another minor issue is that the authors should add an independent conclusion section after the numerical experiments/applications section for the summary, discussion and future research directions of the paper.

---

> ### Author Response · Authors · 2022-08-02
> **Replies to Individual Comments (Reviewer GXHV)**
>
> > The theoretical results only hold under the assumptions that m must be much larger than n and k. However, in the numerical experiments, the IRLS algorithm also performs well under the condition that m is comparable larger than k and n. Hence there is some gap between the theoretical analysis and the numerical experiments. Also the authors should mention in the paper any theoretical results for the case of m is not large enough, i.e., assumption (10) does not hold. Is there any significant results for this special case?
> >
> > > **Reply**: That is a excellent point and we are happy to clarify this further. The gap between the constants of Proposition 3 and the empirical behavior of IRLS is correctly identified. However, Proposition 3 and assumption (10) can be justified as follows. First note that the left hand side of (10) is approximately equal to m-n. The right hand side is of order \Omega(k), ignoring logarithmic factors. Thus (10) amounts to m-n = \Omega(k). Up to logarithmic and constant factors, Proposition 3 is nearly optimal, as the assumption m-n>2k is necessary for unique recovery via L0 minimization (Lines 151-153). In other words, one can improve Proposition 3 and assumption (10) only up to some logarithmic and constant factors. While the gap between (10) and experiments imply that the constant or logarithmic factors can be significant, improving (10) is mathematically more challenging (e.g., we do not know how to dispense with the logarithmic factors). While we are not aware of significantly weaker assumptions than (10) that require a smaller number of observations $m$ and guarantee the stable RSP, and we would like to reference and discuss such a result if it is pointed out.
>
> > Another minor issue is that the authors should add an independent conclusion section after the numerical experiments/applications section for the summary, discussion and future research directions of the paper. Could the authors put the discussion and future work on page 9 of the paper into one independent conclusions section? This can make the paper more clean and clear.
> >
> > > **Reply**: Yes, that's a great point. We do have a short paragraph about summary and future research directions at the end of the main paper (Lines 313-320). In fact, there are several fundamental and important research directions that our paper might inspire and we would like to discuss in more detail than what we currently have at Lines 313-320. We did not expand that paragraph into a full section due to space constraints (please note that every line of the main paper is almost full, except empty lines between paragraphs/sections).
>
> > **Promised Revision**: In our revision, we will discuss more carefully the stringency of assumption (10). We will also create an independent section for summary and discussion of future works. Please let us know if there is anything that we could do to improve the presentation, thanks!

---

> > ### Comment · Reviewer_GXHV · 2022-08-07
> > **Response to Authors**
> >
> > I thank the authors for their detailed response and clarification on each comments. I remained my point of view. Thanks.

---

### Official Review · Reviewer_qErq · 2022-07-11

**Rating:** 6
**Confidence:** 4
**Soundness:** 4 excellent
**Presentation:** 3 good
**Contribution:** 3 good

**Summary:**

This paper studies the classical algorithm of IRLS for solving $\ell_p$ regression for $p \in (0,1]$. The algorithm updates between solving a weighted least squares problem and updating the weights, where the authors also use a dynamically updated smoothing (regularization). Under a stable range space property, which is satisfied by Gaussian matrices, the authors prove that IRLS with $p=1$ converges linearly. Furthermore, for $p<1$, the authors show that the algorithm converges at a superlinear rate provided that the algorithm has good local initialization.

**Questions:**

- What is the point of the $(p,k,\eta)$ RSP when even for the case of $p<1$ you use the $(1,k,\eta)$ RSP?
- Is there a connection between the IRLS algorithm with $p=0$ and a Newton method, which is typical of exhibiting such quadratic rates?
- Does Proposition 3 hold for sub-Gaussian rather than just Gaussian?
- Since the regularization update as well as the reweighting scheme have been considered before, what are the “novel variants of IRLS” considered here? What other notions of IRLS with $p<1$ exist in the literature? Some examples are
    - Mohan, K., & Fazel, M. (2012). Iterative reweighted algorithms for matrix rank minimization. The Journal of Machine Learning Research, 13(1), 3441-3473.
    - "​Lerman, G., & Maunu, T. (2018). Fast, robust and non-convex subspace recovery. Information and Inference: A Journal of the IMA, 7(2), 277-336
- What is the connection, if any, between RSP conditions and the notion of restricted isometry properties?



**Limitations:**

Limitations of the algorithm for $p<1$ are discussed in the example of phase retrieval. Also, the suboptimality with respect to $m$ is discussed in 177-179.

**Strengths And Weaknesses:**

Strengths:
- The paper is very clear and well-written.
- The authors give an improved result for the convergence of IRLS with $p=1$ over [15]. In particular, the result of the authors works under a generic condition and uses a better smoothing update rule.
- The result for superlinear convergence with $p<1$ is interesting, and the result extends to the case of $p=0$, which is the first time I have seen this in the literature.
- The choice of $p<1$ for phase retrieval achieves the theoretical minimum number of samples.

Weaknesses:
- There is not a novel algorithmic twist developed here, and the work is gauged to purely developing theory. How realistic is the assumption in the linear regression without correspondences that only $k$ of the rows are permuted?
- It is unclear how stringent the initialization condition in Theorem 2 is.
- There is not a driving application for this method, and it rather serves as a general-purpose methodology.
- There is no discussion of the choice of $p$ in practice. It seems like $p<1$ is beneficial. However, in the case of phase retrieval, it seems to do poorly in the balanced regime of $|I^+| = |I^-|$.

---

> ### Author Response · Authors · 2022-08-02
> **Replies to Individual Comments (Reviewer qErq)**
>
> > How realistic is the assumption in the linear regression without correspondences that only $k$ of the rows are permuted?
> >
> > > **Reply**: This is an assumption that was proposed in [54]. The justification in [54] is that there exists domain-specific "record linkage" methods that are able to correctly match part of entries of y and rows of A. [54] showed the usage of some record linkage algorithm combined with L1 robust regression on real data. In fact, this record linkage problem and thus linear regression without correspondences is of great interest to US and German governments. E.g., see:
> > >
> > > > https://www.census.gov/topics/research/stat-research/expertise/record-linkage.html
> > > >
> > > > https://papers.ssrn.com/sol3/papers.cfm?abstract_id=3549199
> > >
>
> > There is not a driving application for this method, and it rather serves as a general-purpose methodology.
> >
> > > **Reply**: With this we partly agree. While IRLS is a general-purpose algorithmic framework, we intentionally restricted ourselves to the robust regression setting, and the 3 applications that we considered all admit robust regression formulations, to which our theory directly applies. On the other hand, we are aware of a number of applications of IRLS in computer vision. While these applications involve outlier-robust estimation and the IRLS variants explored here can be applied algorithmically, our theory does not transfer directly to those applications, the reason being that these applications involve optimization over non-convex sets (e.g., Stiefel manifolds, rotations). Global linear rate analysis of IRLS under this extra non-convexity requires more efforts that might not be achieved in a single paper, which is thus left to future work (Line 319). In fact, such a global linear rate analysis for the problem "robust subspace recovery"  is still open, see, e.g.,
> > >
> > > > Table II, https://arxiv.org/pdf/2001.06970.pdf
>
> > What is the point of $(p,k,\eta)$-RSP when even for the case $p<1$ you use the $(1,k,\eta)$-RSP?
> >
> > > **Reply**: Great observation. We will define $(1,k,\eta)$-RSP only.
>
> > Does Proposition 3 hold for sub-Gaussian rather than just Gaussian?
> >
> > > **Reply**: Yes, we implied this at Line 192 and Line 315.
>
> > What is the connection, if any, between RSP conditions and the notion of restricted isometry properties?
> >
> > > **Reply**: This has been precisely discussed in [48,Proposition 3.6]; see also [27].
>
> > **Promised Revision**: The questions/weaknesses pointed out are incredibly useful. We will make changes accordingly (e.g., we will talk about Mohan and Fazel, sub-Gaussian). We might not be able to incorporate the discussion on Newton's method in the main paper due to space constraints.

---

### Author Response · Authors · 2022-08-02
**Replies to Common Concerns**

Here we address two common concerns that are raised by *Reviewer qErq* and *Reviewer yi3J*: (1) the choice of p in practice, (2) the condition of Theorem 2 that defines the local convergence radius of IRLS-p.

> (1) There is no discussion on the choice of p in practice (*Reviewer qErq*). Is p=0.1 typically (approximately) the best choice in all experiments? What if you choose p=0.05? (*Reviewer yi3J*)
>
> > **Reply**: First of all we note that the choice p=0.1 that we used is quite arbitrary. Our theory (Theorem 2) establishes that IRLS exhibits an order of convergence of 2-p, at least locally, which suggests that a smaller choice of p leads to a faster algorithm. This has been also observed empirically as illustrated in Figure 4 of the supplementary material. In particular, Figure 4a shows that, to reach machine accuracy, IRLS-0.1 needs 6 iterations, IRLS-0.5 needs 7 iterations, and IRLS-1 needs roughly 20-30 iterations. While IRLS-0.1 and IRLS-0.5 clearly converge faster than IRLS-1, the advantage of IRLS-0.1 over IRLS-0.5 is not that obvious. From this experiment one might conclude that the speed improvement of IRLS-0.05 or IRLS-0 (see Appendix A.2 for an explanation about IRLS-0) compared to IRLS-0.1 is expected to be not very significant; this is in fact our empirical observation.
> >
> > While *Reviewer qErq* pointed out that IRLS-0.1 seems to do poorly in the balanced regime, we note that, to our knowledge, there is currently no algorithm that performs well in the balanced regime with minimum samples. On the other hand, just like other algorithms, IRLS-0.1 will perform well in the balanced regime if a sufficient number of samples is provided (Lines 270-276).
> >
> > **Promised Revision**: In our revision, we will add an explanation on the choice of p, and will plot a figure that reports the number of iterations that IRLS-p needs to converge, for a larger number of choices of $p\in[0,1]$.

> (2) It is unclear how stringent the initialization condition in Theorem 2 is (*Reviewer qErq*). The radius of the convergence zone (RHS of eq. 13) for the nonconvex IRLS depends on the min over the nonzero residuals. Thus, the radius could be small when k (the number of nonzero elements), n, m are all large (*Reviewer yi3J*).
>
> > **Reply**: That is an interesting and powerful concern. Mathematically, the right-hand side of (13) could be very close to zero and condition (13) might ask for too much. In this sense, the arguments of *Reviewer yi3J* are absolutely correct, and the two reviewers have raised an excellent concern.
> >
> > Yet, we would expect that the residual $|a_i^\top x^* - y_i|$ of every outlier sample $(a_i,y_i)$ to be large; that's what people typically mean by outliers. A natural extra assumption would be $|a_i^\top x^* - y_i|> \delta$ for some constant $\delta>0$ and for every $i\in S^*$. Thus  $\delta$ is a quantity that measures the "outlier-ness" of the data. In that sense, we believe that the right-hand side of (13) is an reasonable term to define the local convergence radius, given also that, unlike for previous results for comparable IRLS algorithms [24], our convergence radius (13) is more well-behaved (Lines 236-244).
> >
> > To give a complete story, one could also consider the case where  $|a_i^\top x^* - y_i|$ is non-zero but very close to zero. In this case, it is more natural to treat  $(a_i,y_i)$ as a "noisy inlier" rather than as an outlier. That is precisely the setting of [15, Section 7.3]. In such a situation, one would instead assume that (1) $S^*$ is the index set corresponding to the largest $k$ entries of $r^*$ in absolute values, (2) the entries of $r^*$ indexed by $S^*$ are large enough, compared to the rest entries, meaning that $S^*$ is the index set for outliers that undergo large corruptions, (3) the remaining entries of $r^*$ are i.i.d. and follow some zero-mean Gaussian distribution with small variance, corresponding to "noisy inliers". If the variance is zero, then this case reduces to the exact $k$-sparse case that we considered in the paper.
> >
> > In fact, we have similar convergence results for this setting with "noisy inliers", but we decided not to present such results in the main paper, for three reasons: (1) it would make the setting and theorems more complicated, obscures the main idea behind IRLS, and hurts the clarity and readership, (2) the proof would be a slight variant of what we have done for the exact sparse case, (3) there would be no space to discuss them in the main paper.
> >
> > **Promised Revision**: Still, it is very important to clarify the convergence radius defined in (13). We will incorporate an improved discussion in our revision.

---

### Author Response · Authors · 2022-08-02
**IRLS versus Newton's method**

Here we discuss an intriguing connection between our IRLS variant and Newton's method (as per a question from *Reviewer qErq*). To fully appreciate this connection, we refer the reviewers to Appendix A, where we elaborate how IRLS works.

> Is there a connection between the IRLS algorithm with $p=0$ and a Newton method, which is typical of exhibiting such quadratic rates?
>
> > **Reply**: That's an amazing question. In short, yes. Details are listed below:
> >
> > - Indeed, it is conceivable to derive a variant of Newton's method for a sum-of-logarithm objective (defined at Line 663 in Appendix A.2). Indeed, if combined with an appropriate smoothing strategy (see for example Section 3 of "A survey of some nonsmooth equations and smoothing Newton methods" by Qi and Sun, In Progress in optimization, 1999.) that Newton's variant might exhibit likewise superlinear or quadratic local convergence rates.
> > - Similarly to Newton's method, IRLS can be considered as a second order algorithm. The difference is that a Newton's method minimizes a *quadratic approximation* of the objective at each iteration, while IRLS minimizes a *quadratic majorizer* of the objective (see Appendix A). The conceptual difference between Newton's method and IRLS becomes most apparent in the non-convex case of p < 1: Due to non-convexity of the smoothed objective $H_{\epsilon}$ (defined at Line 599 in Appendix A.1), there are regions where the Hessian of $H_{\epsilon}$ (which would be directly used in a Newton's method) has negative eigenvalues, whereas the matrix governing the quadratic term of the quadratic majorizing model function $Q_{\epsilon}(\cdot,\mathbf{r})$ (defined at Line 610 in Appendix A.1) used by IRLS has by definition always only _positive_ eigenvalues.
> > - Crucially, for IRLS, as a quadratic majorizer is used, it is _guaranteed_ that the objective value $H_{\epsilon^{(t)}}(\mathbf{r}^{(t)})$ decreases at each iteration, see, e.g., Appendix A. On the other hand, such a monotonuous decrease will not be guaranteed for a variant of Newton's method as the Newton direction might not be a descent direction.
> > - While this connection has not been intensively studied in the literature yet, the ICML 2021 paper [39] explores how precisely IRLS incorporates second-order information: It is argued that the matrix governing the quadratic model function $Q_{\epsilon}(\cdot,\mathbf{r})$ is such that negative eigenvalues of the Hessian $\nabla^2 H_{\epsilon^{(t)}}(\mathbf{r}^{(t)})$ are "flipped" to positive sign, allowing IRLS to exhibit similarities to a "saddle-escaping" modification of a Newton's method specifically designed for non-convex objectives; the latter had been analyzed in the paper "A Newton-based method for nonconvex optimization with fast evasion of saddle points" Paternain, Mokhtari and Ribeiro, SIAM Journal on Optimization, 2019.

---

### Author Response · Authors · 2022-08-02
**The Algorithmic Twist**

In this post, we reply to the concern of *Reviewer qErq* on the novelty of the algorithms.

> There is not a novel algorithmic twist developed here.
>
> > **Reply**: While we agree very much that the main contribution of our paper lies in its theoretical results, we would like to respectfully point out that we do propose an algorithmic twist.  First of all, the basic IRLS-p framework (with fixed smoothing parameter) is known, and the update rule (7) that we employ was proposed by [25] in the context of compressed sensing with p=1. The algorithmic novelty is that we apply the IRLS framework with rule (7) to the context of Lp robust regression, for $p<1$. We haven't seen Algorithm 1 with update rule (7)  in its precise form in the literature. While the twist here appears as simple as combining a few known ideas, it turns out to be very crucial, as it allows IRLS to work pretty well particularly for small p (e.g., $p\in[0,0.1]$), for which we provide theory support.
> >
> > For IRLS-$p$ with $p\in[0,1]$, this update rule is critical for obtaining a larger local convergence radius as well as a competitive empirical performance. In particular, the improvement discussed in Lines 236-244 is enabled by the update rule (7), and the improved empirical behavior is explored in Figure 1b (see also our reply to Reviewer yi3j discussing the different smoothing parameter rules).

> Question 1: Since the regularization update as well as the reweighting scheme have been considered before, what are the “novel variants of IRLS” considered here? Question 2: What other notions of IRLS with $p<1$ exist in the literature? Some examples are
> - Mohan, K., & Fazel, M. (2012). Iterative reweighted algorithms for matrix rank minimization. The Journal of Machine Learning Research, 13(1), 3441-3473.
> - Lerman, G., & Maunu, T. (2018). Fast, robust and non-convex subspace recovery. Information and Inference: A Journal of the IMA, 7(2), 277-336.
> > **Reply 1** : Thank you for this important question. As clarified above, by "novel variants of IRLS" we are referring to the precise formulation of Algorithm 1 using smoothing parameter update rule (7), which has not been considered in the context of IRLS for Lp robust regression, and which makes IRLS competitive in the non-convex regime of $p<1$, including $p=0$.
> >
> > **Reply 2**: Thank you for pointing out the two important references! There is indeed more literature about IRLS with $p<1$ for similar problems which we will add in the discussion of Section 3.3.
> >
> > - The paper [Mohan and Fazel 2012] generalized the IRLS framework to the problem of low-rank matrix recovery, which involves an objective that is _not_ separable, which is an additional algorithmic challenge not present for compressed sensing or robust regression. Their weight matrix choice extends to the case of $p < 1$, but it does not provide any local convergence result of the type of our Theorem 2. Variants of the smoothing parameter update rule (5) are explored in their numerical experiments, but do not lead to empirically observed superlinear convergence. We will add a reference to this important work. The references [37] and [39] improve on [Mohan and Fazel 2012] since their IRLS variants do lead to superlinear convergence, which is also established theoretically. The smoothing parameter update rules of [37] and [39] are different from both (6) and (7) and are adapted to the specificity of the matrix recovery problem. We omitted a detailed discussion of these three works as they study the optimization of nonseparable objectives.
> > -  Lerman and Maunu [71] analyzed an IRLS method for the important problem of robust subspace recovery (which is different from robust regression and somewhat more challenging).  In their paper [71] they work with a fixed smoothing parameter and thus does not yield convergence to a true solution of the original problem (Line 99). The convergence rates that are achieved, first, refer instead to the convergence of the IRLS sequence to a _stationary point of the $\epsilon$-smoothed objective_ and, second, are globally sublinear and locally linear independently from $p$, whereas our result is globally linear ($p=1$) and locally _superlinear_ (for $p < 1$).

---

### Author Response · Authors · 2022-08-02
**Summary on Reviews**

We thank all the reviewers for their intellectual inputs and expert comments, which are concise, constructive, and to the point. In fact, we are a bit surprised to receive such high-quality reviews, given the current scale of NeurIPS and the amount of papers that each reviewer receives.

Clarity is what we consider as the most important factor in a paper with mathematical contents, and we are deeply grateful to all reviewers as they acknowledged our efforts in writing and presentation. Equally important is the technical/theoretical novelty and contribution, which has been commented by all reviewers as another strength of the paper. Also, we are glad that *Reviewer qErq* finds that the paper/method "serves as a general-purpose methodology", as this implies that our study has potential impacts to multiple fields, including (1) the applications explored in Section 4, to which our theory directly applies, (2) even more applications in computer vision, to which the algorithmic idea of IRLS applies (Lines 319). Overall, we thank all reviewers for their positive views and positive ratings on the paper.

That said, there are still several points to clarify, which have been listed as weaknesses or questions in the reviews. In the rest of our reply, we will focus on justifying the weaknesses and answering questions, and we will propose several changes that respect reviewers' comments. For a camera-ready version, we will be able to incorporate the valuable feedback we received within the limits of the extra page if the paper is accepted. We wish our reply would alleviate the concerns of reviewers and help to further improve a reader's understanding of the paper and the proposed IRLS algorithm.

Finally, it is great that there is an Author-Reviewer Discussion phase. Please let us know if there are any concerns that we failed to fully address in our rebuttal, if there are any new issues that deserve discussion, or if there is anything that we could do and might improve the paper.

---

### Meta-Review · Area_Chair_JdHB · 2022-08-23

**Recommendation:** Accept
**Confidence:** Certain

**Metareview:**

Thank you for your submission to NeurIPS. The reviewers unanimously found the contributions to be solid, and the paper to be clear and well-written. All three reviewers unanimously recommend accepting the paper. One reviewer remarks that "it is actually interesting to see that the smoothing parameter plays a critical rule in the convergence of IRLS, and I believe that it has been ignored by many practitioners in this area." Please incorporate reviewer feedback in preparing the camera ready version.

**Award:**

No

---

### Decision · Program_Chairs · 2022-09-14

Accept